



# Satellite observed indications of aerosol effects on warm cloud properties over Yangtze River Delta of China

Yuqin Liu[1, 2, 3], Gerrit de Leeuw[1,3, 4], Veli-Matti Kerminen[3], Jiahua Zhang[*1], Putian Zhou[3], Wei Nie[5], Ximeng Qi[5], Juan Hong[3], Yonghong Wang[3], Aijun Ding[5], Huadong Guo[1], Olaf Krüger[3], Markku Kulmala[3], Tuukka. Petäjä[3]

[1]Institute of Remote Sensing and Digital Earth, Chinese Academy of Sciences, Beijing, China

[2]University of Chinese Academy of Sciences, Beijing, China

[3]Department of Physics, P.O. Box 64, 00014 University of Helsinki, Helsinki, Finland

[4]Finish Meteorological Institute, Climate Change Unit, P.O. Box 503, 00101 Helsinki, Finland

[5]Institute for Climate and Global Change Research & School of Atmospheric Sciences, Nanjing University, 210023 Nanjing, China

*Correspondence to*: J.H. ZHANG (jhzhang@ceode.ac.cn)

**Abstract.** Aerosol effects on summertime low warm clouds over the Yangtze River Delta (YRD) are examined using co-located MODIS, CALIOP and CloudSat observations. By taking the vertical locations of aerosol and cloud layers into account, we use simultaneously observed aerosol and cloud data to investigate relationships between cloud properties and the amount of aerosol particles (using aerosol optical depth, AOD, as a proxy). Also, we investigate the impact of aerosol types on the variation of cloud properties with AOD. Finally, we explore how meteorological conditions affect these relationships using ERA Interim Reanalysis data. This study shows that the relation between cloud droplet effective radius (CDR) and AOD depends on the aerosol abundance, with a different behaviour for low and high AOD (i.e. AOD <> 0.3). Cloud fraction (CF) is found to be little dependent on the AOD when aerosol and cloud physically interact, but has a positive relation in case of well-separated clouds. Cloud optical Thickness (COT) is found to decrease when AOD increases, which may be due to radiative effects and retrieval artefacts caused by absorbing aerosol. Conversely, cloud top pressure (CTP) tends to increase with elevated AOD, indicating that the aerosol is not always prone to expand the vertical extension. Furthermore, separation of cases with either polluted dust or smoke aerosol shows that COT and CF are smaller for clouds mixed with smoke aerosol which is ascribed to the higher absorption efficiency of smoke than dust. The variation of cloud properties with AOD is analysed for different values of relative humidity (RH) and boundary layer thermodynamic and dynamic conditions, showing that high relative humidity and upward motion of air parcels can enhance the strength of aerosol-cloud interaction, especially pronounced in heavily polluted conditions than in moderately polluted conditions. Meteorological conditions play a weak role in the COT-AOD and CTP-AOD relationships throughout the range of AOD. Overall, the interpretation of the observed relationships between cloud properties and AOD requires that ambient environmental conditions are considered in addition to the aerosol and cloud parameters.

## 1. Introduction

Aerosol impacts on clouds and precipitation have been reported to introduce the largest uncertainty in quantifying the anthropogenic contribution to climate change (Rosenfeld, 2000; Twomey, 1974;



Gryspeerdt and Partridge, 2014; Kaufman et al., 2012). Atmospheric aerosol particles have been recognized to have two effects on Earth's climate. First, they can directly alter the energy balance due to scattering and absorption of incoming solar radiation (e.g. McCormick and Ludwig, 1967), and second, they can act as cloud condensation nuclei (CCN) and thus modify the cloud microphysical properties and

lifetime as well as precipitation (Ramanathan et al., 2001; Krüger and Grassl, 2011). The effects of aerosol-induced changes of cloud properties on the radiation budget are collectively referred to as aerosol indirect effect (AIE). The study presented here is confined to aerosol-cloud interaction (ACI) using satellite data.

The activation of aerosol particles to CCN, or more specifically the number concentration of CCN, is a

direct link between aerosols and clouds and the aerosol activation efficiency is a key aerosol property affecting aerosol-cloud interaction. For a given constant cloud liquid water path (CWP), an increased aerosol concentration is expected to lead to smaller and more numerous cloud droplets, resulting in an increase of cloud albedo. This process, termed as the "first AIE" or "Twomey's effect", may lead to a net cooling of climate (Twomey, 1974; Feingold et al., 2003). The reduced cloud droplet effective radius

(CDR) also suppresses precipitation and can consequently increase cloud lifetime, thus maintaining a larger liquid water path, with a possible further increase in the cloud optical thickness (COT) and cloud reflectance. This process, described as the "second AIE", may further influence the cloud fraction (CF) (Albrecht, 1989; Feingold et al., 2001). The interaction mechanisms between aerosols and clouds remain among the most uncertain processes in the global climate system in spite of a large number of studies

make through both observations (Platnick et al., 2003; Koren et al., 2005) and models (Suzuki et al., 2004; Quaas et al., 2009; Sena et al., 2016).

In order to better understand aerosol indirect effects, we resorted to statistical analysis of satellite observations. By virtue of their large coverage and high spatial and temporal resolution, satellite-borne instruments have become a promising observational tool in studying aerosol-cloud interactions. Previous

studies using a large amount of satellite data and / or multiple satellite instruments have shown that aerosol particles can affect cloud properties significantly (Krüger and Grassl, 2002; Costantino and Bréon, 2010; Menon et al., 2008; Sporre et al., 2014). Satellite measurements suggest that the CDR tends to decrease with an increasing aerosol loading, which is consistent with Twomey's theory (Matheson et al., 2005; Meskhidze and Nenes, 2010; Koren et al., 2005). However, also positive correlations between

CDR and aerosol optical depth (AOD) have been found in some study areas, from both observations and





models, especially over land (Feingold et al., 2001; Grandey and Stier, 2010; Yuan et al., 2008). Different behaviours of CDR as function of AOD for different AOD regimes (low/high) have been observed by, e.g., Tang et al. (2014) and Wang et al. (2015). Feingold et al. (2001) concluded that there are three kinds of CDR responses to aerosol enhancement: the CDR decreases with increasing aerosol

loading followed by (1) a saturation of the value of CDR in response to high AOD, (2) a decrease in the CDR with further increasing AOD due to suppression of cloud water vapour supersaturation caused by abundant large particles, or (3) an increase in CDR with further increases in AOD due to an intense competition for vapour which evaporates the smallest droplets. Likewise, the aerosol impact on COT is still poorly quantified. Costantino and Bréon (2013) reported that the relationship between AOD and

COT, which can be either positive or negative, depends on the balance between the simultaneous CDR increase and CWP decrease when AOD increases. With regard to the impact of aerosols on the cloud life cycle, it is of great importance to explore the relationship between the aerosol loading and cloud fraction, because the cloud fraction is highly associated with other cloud properties and has a large effect on radiation (Gryspeerdt et al., 2016). Kaufman (2006) and Koren (2008) reported an increase in the cloud

cover with an increasing aerosol loading, followed by an inverse pattern due to the absorption efficiency of aerosol. This brief summary shows that the aerosol effect on cloud properties and the magnitude of this effect are still very unclear.

Aerosol and cloud properties may have different vertical distributions and may actually not physically interact. Costantino and Breon (2013) and Jones et al. (2009) found that the aerosol indirect effect is

stronger for well-mixed clouds than for well-separated clouds (in well-mixed clouds aerosol and cloud layers are physically interacting, as further explained in Section 2) when using MODIS data. These observations show that it is important to consider the relative altitudes of aerosol and cloud layers when estimating the aerosol indirect effects. In addition, local differences in aerosol populations and cloud regimes may have a strong effect on ACI (Sinha et al., 2003; Small et al., 2011; Kaufman et al., 2005b).

Yuan et al. (2008) proposed that the chemical composition of aerosol particles may play a role in determining the relationship between AOD and CDR. Meteorology can affect the interaction between aerosol and cloud, which usually further complicates ACI (Koren et al., 2010; Reutter et al., 2009; Loeb and Schuster, 2008; Su et al., 2010; Stathopoulos et al., 2017). As a consequence, the widely varying estimates of the aerosol impact on cloud parameters, either positive or negative, depend on factors like

the aerosol size distribution and chemical composition, cloud regime and local meteorological conditions.





Therefore, the dataset used in this study contains not only aerosol and cloud properties derived from MODIS, CALIOP and CloudSat, but also the meteorological parameters collected from the daily ERA Interim Reanalysis data.

The Yangtze River Delta (YRD) is characterized by a variable aerosol composition and increasing

aerosol concentration in the last two decades (Ding et al., 2013a; Qi et al., 2015). Using multi-sensor retrievals, this study aims to systematically examine the response of warm cloud parameters (CDR, CF, COT and CTP) to the increase in the aerosol loading, where AOD is used as a proxy for CCN number concentration (Andreae, 2009; Kourtidis, et al., 2015). New insight into this topic results from our focus on a systematic understanding of ACI from three perspectives: (1) well-mixed and well-separated clouds,

(2) aerosol effects on properties of well-mixed clouds, (3) well-mixed clouds under different meteorological conditions. The paper is organized as follows: section 2 describes the used datasets and data processing and presents the main analysis conducted to explore the aerosol cloud interaction. Section 3 starts with a general description of cloud properties and the effect of aerosol loading on the relations between them, followed by a description of aerosol effects on cloud properties (CDR, CF, COT

and CTP). In the latter we discriminate between well-separated and well-mixed clouds. The focus will be on well-mixed clouds where ACI takes place, and aerosol types and meteorological factors are considered to better understand the possible mechanisms. Overall conclusions and discussions are presented in section 4.

## 2. Methods

### 2.1 Description of region interest

In this study, the Yangtze River Delta (YRD), covering the area 27 °N-34 °N and 115 °E-122 °E, has been chosen to study the aerosol-induced variability of micro- and macrophysical properties of low warm clouds in the summer (June, July and August) during four consecutive years (2007-2010). The YRD region is chosen because it is representative for continental East Asian subtropical climate. The marine

monsoon subtropical climate for YRD is characterized by hot and humid summers and cool dry winters (Sundström et al., 2008; Zhang et al., 2010). The mean temperature in summer is about 27-28℃. Mean annual precipitation ranges from 1 000 to 1 400 mm and most precipitation occurs in spring and summer (Zhang et al., 2010; Cao et al., 2016).



The population density in the YRD is very high with intensive human activities in the region contributing to a very variable and complex aerosol composition. The YRD has been reported to be a major source region of both black carbon and sulfate in the summertime (Wang, et al., 2014; Andersson, et al., 2015). In addition, there are other aerosol sources such as dust emissions, which make the interactions between aerosols and clouds complicated (Nie et al., 2014). The continental area of interest is characterized by a high level of anthropogenic emissions and is well suited for research related to the indirect effects of aerosols on cloud micro- and macro-physical properties.

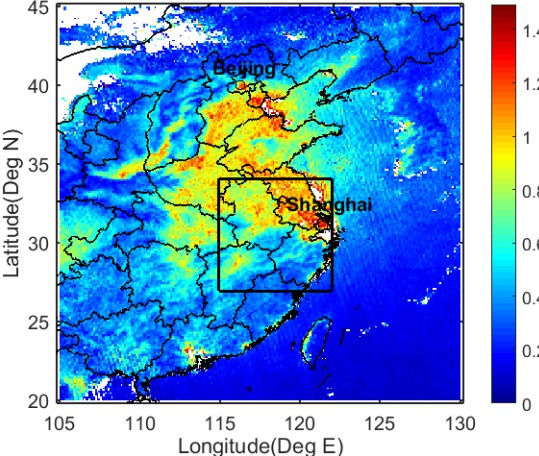

Figure 1. Map of averaged MODIS/AQUA level 2 AOD in the summertime (JJA) during the period from 2007 to 2010. The black rectangle (27 °N-34 °N and 115 °E-122 °E) indicates the Yangtze River Delta (YRD).

**2.2 Data sources**

The MODIS sensors, onboard the Terra and Aqua satellites, have a swath width of ~2300 km and multi-band spectral coverage (King et al., 2003). The MODIS/Aqua overpass time for the study area is around 13:30 local time, when continental warm clouds are likely to be well developed. Therefore MODIS/Aqua is selected as a data source to explore the ACI over this area. In this work, we use the MODIS Collection 5.1 AOD product (MOD04) derived from cloud-free pixels (resolution 500 m at nadir) and aggregated to a resolution of 10 km×10 km (Remer et al., 2005; Levy et al., 2010). The AOD over land is retrieved using three MODIS channels: 0.47, 0.66 and 2.13 μm (Remer et al., 2005). Cloud properties are retrieved using six spectral channels (King et al., 1998) at visible and near infrared wavelengths (i.e., 0.66, 0.86, 1.24, 1.64, 2.12 and 3.75 μm). Here, we use the AOD as a proxy for aerosol





burden in our aerosol-cloud interaction analysis. The cloud properties used in this study, CDR, CWP, COT, cloud top pressure (CTP) and cloud phase infrared (CPI), are obtained from the Level 2 cloud product (MYD06) (King et al., 2003). Both the products MOD04 and MYD06 are in good agreement with ground-based remote sensing data (Levy et al., 2010; Platnick et al., 2003). More detailed

information on algorithms for the aerosol and cloud retrievals is provided at http://modis-atmos.gsfc.nasa.gov.

Along with the Aqua and Terra satellites, also CALIPSO and CloudSat are flying in the so-called "A-train" constellation together with other NASA satellites (Stephens et al., 2002). CloudSat was the first mission to fly the first satellite-based millimeter-wavelength cloud radar to detect the vertical

information on different sized cloud droplets (Im et al., 2005). The CPR (Cloud Profiling Radar), which is carried on the CloudSat, is able to penetrate optically thick clouds and detect weak precipitating particles (Wang et al., 2013). In the present study we utilize the dataset of CloudLayerBase and CloudLayerTop from 2B-CLDCLASS-LIDAR, the latest version (R04) of CloudSat standard data products. The data are provided in the CPR spatial grid with vertical and horizontal resolutions of

approximately 480 m and 1.4×1.8 km, respectively. CALIOP (Cloud-Aerosol Lidar with Orthogonal Polarization) on board CALIPSO is the first space-borne near-nadir polarization lidar optimised for aerosol and cloud measurements (Winker et al., 2003). It is sensitive to optically thin clouds which could be missed by CPR (Wang et al., 2013). The datasets of Layer_Base_Altitude and Layer_Top_Altitude retrieved from the CALIOP level-2 aerosol layers product (05kmALay) are used in the present study. Its

footprint is very narrow, with a laser pulse diameter of 70 m on the ground. Combining CloudSat and CALIPSO observations has provided new insights into the vertical structure and microphysical properties of clouds (Matrosov, 2007).

The daily temperature at the 1000 hPa and 700 hPa levels, relative humidity at the 950 hPa level and pressure vertical velocity at the 750 hPa level are obtained from ERA Interim Reanalysis data. The daily

ERA Interim Reanalysis contains global meteorological conditions with 0.125 °×0.125 ° grids and a 37 level vertical resolution (1000-1 hPa) every six hours (00:00, 06:00, 12:00, 18:00 UTC) (http://apps.ecmwf.int/datasets/data/interim-full-daily/). The reanalysis data are used for the closest collocation with the satellite overpass time over the study area.

Table 1. Level 2 MODIS, CALIOP, CALIOP/CPR and ERA Interim products used to characterize aerosol and cloud

properties.



| Product | Dataset | Horizontal resolution | Data source |
|---|---|---|---|
| Aerosol(MYD04 Level 2 Collection 5) | Optical_Depth_Land_And_Ocean | 10 km | MODIS |
| Cloud(MYD06 Level 2 Collection 5) | Cloud_Effective_Radius | 1 km | |
| | Cloud_Water_Path | 1 km | |
| | Cloud_Phase_Infrared_Day | 5 km | |
| | Cloud_TOP_Pressure_Day | 5 km | |
| | Cloud_Fraction_Day | 5 km | |
| | Cloud_Optical_Thickness | 1 km | |
| Cloud(2B-CLDCLASS-LIDAR) | CloudLayerBase | 2.5 km | CALIOP/CPR |
| | CloudLayerTop | 2.5 km | |
| Aerosol(05kmALay) | Layer_Top_Altitude | 5 km | CALIOP |
| | Layer_Base_Altitude | 5 km | |
| | Cloud_Aerosol_Discrimintion | 5 km | |
| Aerosol(VFM) | Feature_Classification_Flags | 5 km | |
| ERA Interim | Temperature (700hPa, 1000hPa) | 0.125° | ECMWF |
| | Relative humidity (950hPa) | 0.125° | |
| | Pressure vertical velocity (750hPa) | 0.125° | |

### 2.3 Data processing

The MODIS/AQUA, CALIOP/CALIPSO and CPR/CLOUDSAT satellites are part of the A-Train constellation and observe the same scene on Earth within one to two minutes (Stephens et al., 2002).

Therefore, time-coincidence of retrievals is assured when the datasets are extracted for the same date. Meteorological properties retrieved from the 06:00 UTC ERA Interim datasets are used here as the "A-train" satellites constellation overpasses the region of interest at about 13:30 local time (05:30 UTC). We aggregate CDR, COT and CWP (1 km ×1 km) to a resolution of 5 km ×5km to match the along-track resolution of CALIOP (5 km ×5 km), while CTP, CF and CPI are directly applied for the analysis since

all of them are at a 5 km ×5km spatial resolution.

Aerosol properties are only retrieved for strictly cloud-free pixels as determined by application of a cloud-detection scheme. However, cloud detection schemes are not perfect and some residual clouds may remain undetected resulting in high AOD (Kaufman et al., 2005b). Another potential source of error could be the misclassification of high AOD areas, such as in the presence of desert dust or very high

pollution levels, as clouds. To reduce a possible over-estimation of AOD, cases with AOD greater than



1.5 are excluded from further analysis. In this paper, we focus on warm clouds with CTP smaller than 700 hPa and CWP lower than 200 g m-2, as most aerosols exist in the lower troposphere (Michibata et al. 2014). In addition, only cases with CPI = 1 (liquid water cloud) are included. When CALIPSO detected the presence of aerosol, we average the MODIS aerosol retrievals within a radius of 50 km from the CALIPSO target. Likewise, we average the MODIS cloud retrievals within a radius of 5 km from the CALIPSO target. For meteorological properties, we chose the value of the footprint that is nearest to the CALIPSO target. MODIS, CALIOP, and CPR datasets are listed in Table 1.

A quantitative relationship between aerosol optical depth and cloud properties has been documented in many previous studies (Sporre et al., 2014; Meskhidze and Nenes, 2010; Koren et al., 2005). However, the relative vertical positions of aerosol and cloud layers contribute to the uncertainty in this relationship. According to the method by Costantino and Breon (2013), aerosol and cloud layers are considered physically interacting (well mixed) when the vertical distance between bottom (top) of the aerosol layer and the top (bottom) of a cloud layer is smaller than 100 m. Coincident samples with a vertical distance larger than 750 m are assumed to be "well separated". Coincident samples with distance between 100 and 750 m are defined as "uncertain". The uncertain cases, as identified using the information from CloudSat, are excluded from a further analysis in this study. Cloud types are identified as single-, double- and multi-layer clouds using the cloud layer information at each point. Single-, double- and multi-layer cloud samples account for 51 %, 32% and 15 % of the total samples, respectively. Using the highest occurrence frequency (OF) of aerosol type below 2.5 km altitude at each point, the aerosol type of highest OF is defined following the Feature_Classification_Flags derived from CALIOP.

Meteorological and aerosol impacts on cloud macrophysics and microphysics are found to be tightly intermingled (Stevens and Feingold, 2009). In an attempt to isolate aerosol effects, the meteorological effects on clouds are explored in a statistical sense. Meteorological properties used here include relative humidity (RH), lower tropospheric stability (LTS) and pressure vertical velocity (PVV). LTS is defined as the difference in potential temperature between the free troposphere (700hpa) and the surface, which is representative of typical thermodynamic conditions (Klein and Hartmanm, 1993). RH, LTS and PVV have been suggested to affect aerosol and cloud interaction (Gryspeerdt and Partridge, 2014; Small et al., 2011). A positive LTS is associated with a stable atmosphere in which vertical mixing is prohibited; negative PVV indicates upward motion of air parcels and vertical mixing.





### 3. Results and Discussions

### 3.1 Overall cloud characteristics

### 3.1.1 Variation of COT and CWP with CDR

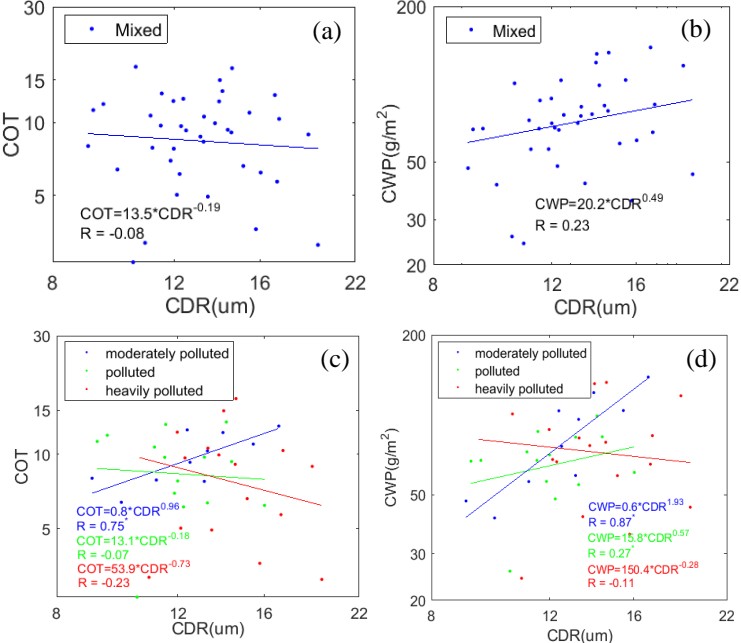

Figure 2. Scatterplots and least square fits of cloud parameters in well-mixed aerosol-cloud layers: (a) COT versus CDR and (b) CWP versus CDR for all data, (c) COT versus CDR and (d) CWP versus CDR for data stratified by moderately polluted (in blue), polluted (in green) and heavily polluted (in red) atmospheric conditions. Here moderately polluted refers to AOD <0.3, polluted refers to 0.3<AOD < 0.6 and heavily polluted refers to AOD>0.6.

The overall statistical associations between the cloud parameters used in this study are derived from the scatterplots shown in Figure 2. All CDR, COT, CTP, CWP data shown in Figure 2 (and later figures) are averaged over AOD bins, from 0.05 to 1.5 by a step of 0.02 on a log-log scale. Student's t-test is used to examine the significance of the difference. * denotes statistically significant (p<0.05).

Figure 2(a) shows a scatterplot of COT vs CDR for well-mixed clouds. The correlation between these parameters is negative but weak, with a correlation coefficient equal to -0.08. Figure 2(c) shows the same data but distinction is made for data points with AOD < 0.3 (moderately polluted), 0.3<AOD<0.6 (polluted) and AOD > 0.6 (heavily polluted) conditions. For this dataset, COT increases with an increasing CDR at moderately polluted conditions. In contrast, for heavily polluted conditions COT





shows a decrease with an increasing CDR albeit with a low significance as indicated by the small

correlation coefficient R. This may indicate the existence of intense competition between the aerosol

particles for water vapour where the larger droplets are more prone for condensation of water vapour than

smaller ones, and thus grow to larger sizes resulting in a shift of the droplet spectrum to larger sizes

resulting in an increase of CDR and decrease of COT (Wang et al., 2015). The data for the three different

AOD cases show that the relationship between CDR and COT is not unique and depends on the aerosol

abundance. Figure 2(b) shows a positive but weak correlation between CWP and CDR for well-mixed

cloud layers, with a correlation coefficient equal to 0.23. Likewise, Figure 2(d) shows the same data as

Figure 2(b) but distinction is made for different AOD regimes as in Fig 2c. We see that at moderately

polluted conditions CWP clearly increases with elevated CDR, while at polluted and heavily polluted

conditions the variation of CWP with increasing CDR is much weaker and even shows a negative

relation. In both cases, the correlations are moderate to high for moderately polluted conditions but rather

weak or even non-existent for polluted and heavily polluted conditions.

### 3.1.2 Variation in COT and CWP with cloud top height

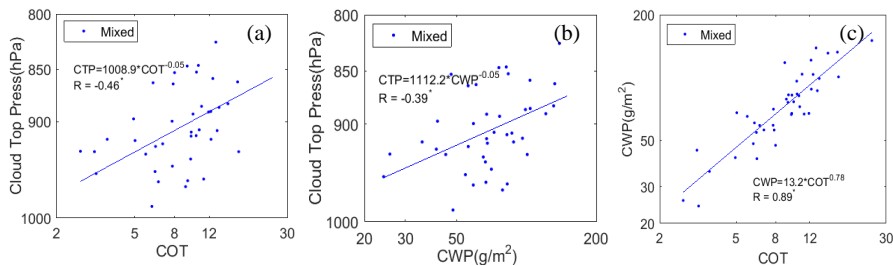

Figure 3. Scatterplots and least square fits of cloud parameters in well-mixed aerosol cloud layers for all data: (a)
CTP versus COT, (b) CTP versus CWP, (c) CWP and COT.

The CTP is generally used as a measure of cloud top height (CTH), with higher CTP implying a lower

CTP. Figure 3(a) shows a positive correlation between CTP and COT, implying the occurrence of higher

clouds with an increasing COT, which is consistent with the general understanding of aerosol-cloud

interactions. Note that here and in the following figures, CTP is plotted along the vertical axis from high

to low, i.e. decreasing CTP indicates increasing CTH, and positive correlations between CTP and other

cloud parameters indicate that an increase in these parameters corresponds to a higher CTH. Figure 3(b)

shows a positive correlation between CTP and CWP, which again implies that clouds are higher as CWP





increases. The explanation for this observation is that clouds grow in the vertical and more drizzle is

produced, so that the cloud liquid water path becomes larger (Gao et al., 2014). Figure 3(c) shows the

relation between CWP and COT. The CWP increases with the increase of COT, which is basically in

good agreement with the aerosol second indirect effect hypothesis that the precipitation suppression can

increase CWP and possibly further increase COT.

### 3.2 Difference between separated and mixed conditions

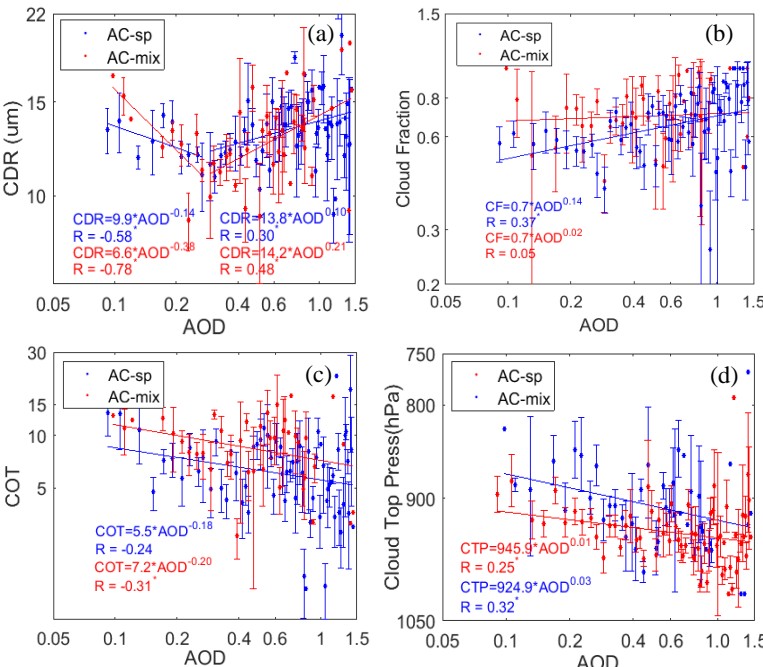

Figure 4. Scatterplots and least square fits of cloud parameters versus AOD over YRD on log-log scale for cases of
separated (blue) and mixed (red) aerosol-cloud layers, (a) CDR versus AOD, (b) CF versus AOD, (c) COT versus
AOD and (d) CTP versus AOD. Error bars represent the confidence level of the mean cloud parameters' value for
each AOD bin, i.e. the statistical uncertainties, expressed as $\sigma/(n-2)$ , where n is the number of cases within the AOD
bin and $\sigma$ is the standard deviation of cloud properties.

Figure 4 shows relations between cloud parameters (CDR, CF, COT, CTP) and AOD for both separated

and mixed conditions. The strength of the interaction between cloud properties and AOD is quantified

here as the slope of the line describing the relation between a cloud parameters and AOD, on a log-log

scale, as obtained by linear regression. In figure 4(a), CDR shows a negative relation with AOD in

moderate polluted conditions (AOD < 0.3), which is in good agreement with Twomey's theory (Twomey,



1977). We note that, due to the limited number of data points in the dataset with AOD < 0.3, the present

work does not allow selecting conditions with a constant CWP. Following, e.g., Costantino and Breon

(2010; 2013) and Wang (2015) we use all available data together. The results in Figure 4a potentially

show that the relation between AOD and CDR is ~3 times stronger for the mixed layers than for

separated layers. In polluted and heavily polluted conditions (AOD > 0.3), however, CDR increases with

increasing AOD, suggesting some sort of saturation in aerosol-cloud interactions when AOD approaches

0.3. This value for the tipping point (0.3) is close to the value of 0.4 reported by Feingold et al. (2001). As

discussed earlier, Feingold et al. (2001) proposed three primary responses of CDR to the aerosol loading.

We consider the fact that CDR increases with an increase in AOD when AOD loading exceeds 0.3 as

"anti-Twomey effect". The positive relation between CDR and AOD may be similar to the that described

by Feingold et al. (2001), case 3 (see above), i.e. due to intense vapour competition the smaller droplets

evaporate as the number of particles continues to increase. It may also be that only a subset of aerosol

particles is activated when not enough vapor is available, and once activated they continue to grow faster,

thus preventing water vapor from condensing onto smaller aerosol particles that are less susceptible to

activation, resulting in the increase of CDR.

Figure 4a also shows that in well-separated cloud layers CDR varies much less with AOD irrespective of

whether the AOD is relatively low or high. Such a weaker variation can be attributed to the fact that no

aerosols are subjected to cloud microphysical process since there are no physical interactions between

aerosol and cloud layers.

Figure 4(b) shows that when aerosol and cloud layers can physically interact, the variation of CF with

increasing AOD is not clear, with no specific dependence on AOD. This outcome is not in agreement

with the findings of Koren et al. (2008) and Small et al. (2011). Conversely, CF shows an increasing

pattern with an increasing AOD in well-separated cloud layers. This increase might be due to absorbing

aerosols interacting with incoming solar radiation above the cloud layer (Costantino and Bréon, 2013). In

this process, absorbing aerosols above cloud top may heat the aerosol layer and cool the surface, thereby

stabilizing the boundary layer and maintaining a moist boundary layer. In addition, scattering aerosol

reduces the amount of solar light reaching the surface. This combination of two effects suppresses cloud

vertical development and increase the low cloud cover.

The COT has a negative correlation with AOD in both conditions, as shown in Figure 4(c). There are two

effects that may contribute to this negative relationship. On one hand, the evaporation of cloud droplets



caused by locally absorbing aerosol makes clouds thinner, which is a radiative effect. On the other hand, the presence of absorbing aerosol may influence the satellite-retrieved COT because it can absorb radiation and thus reduce the cloud reflectance measured by the sensors on the satellite (Meyer et al., 2013; Li et al., 2014; Meyer et al., 2015; Hoeve et al., 2011). Meyer et al. (2013) reported that adjusting

for above-cloud aerosol attenuation can increase the retrieved regional mean COT by roughly 18% for polluted marine boundary layer clouds. Li et al. (2014) also found that, due to absorbing aerosols in the heart of the Yangtze Delta region, satellite observations tend to underestimate COT. The radiative effect and retrieval uncertainty could be the important factors for the decrease of COT with increasing AOD, as suggested by Hoeve et al. (2011) and Alam et al. (2014). These authors reported similar results on the

decrease of COT with increasing AOD, which may result from the measured reflectance from a cloud top at wavelengths in the visible being smaller than expected due to absorbing aerosols.

The relationship between CTP and AOD has been plotted in Figure 4(d). There is a positive correlation between CTP and AOD, which is contradicting the general understanding that high aerosol concentration will result in an increase of cloud lifetime and higher cloud top. The positive relation between CTP and

15 AOD has an implication that higher aerosol abundance is not always accompanied by smaller cloud top pressure. This suggests that the primary effect of aerosol is not always to produce taller and more convective clouds in some cases (Rennó et al., 2013).




### 3.3 Case of mixed aerosol-cloud layers

### 3.3.1 ACI for single-layer mixed clouds

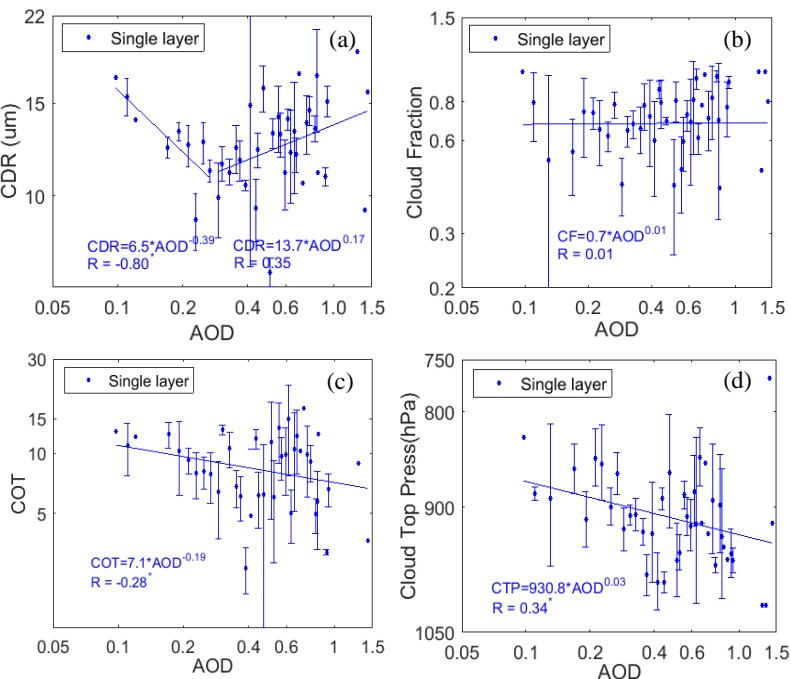

Figure 5. Scatterplots and least square fits of cloud parameters versus AOD over YRD on log-log scale for mixed
aerosol-single layer clouds, (a) CDR versus AOD, (b) CF versus AOD, (c) COT versus AOD and (d) CTP versus
AOD. The error bars indicate the statistical uncertainties as in Fig. 4.

Well-mixed clouds show a stronger relation between aerosol and cloud properties than well-separated
clouds, as shown above. From here on, we will focus on potential aerosol indirect effects on well-mixed
warm clouds as defined above. Relations between CDR, CF, COT and CTP with AOD will be explored
in this section. Figure 5 shows the variation of single layer cloud properties with AOD when aerosol and
cloud layers are mixed. The relation between CDR and AOD changes from negative for AOD < 0.3 to
positive for AOD > 0.3 (Figure 5a). Figure 5(b) shows the low CF sensitivity to aerosol increase
throughout the range of AOD. Figures 5(c) shows that COT is negatively associated with increasing
values of AOD. In contrast, CTP decreases with increasing AOD (Figure 5d), i.e. cloud top height
increases. In general, the characteristics for cases of mixed aerosol-single layer warm clouds (Figure 5)
are quite similar to the case of mixed aerosol-warm clouds (Figure 4).





### 3.3.2 Influence of aerosol type on ACI

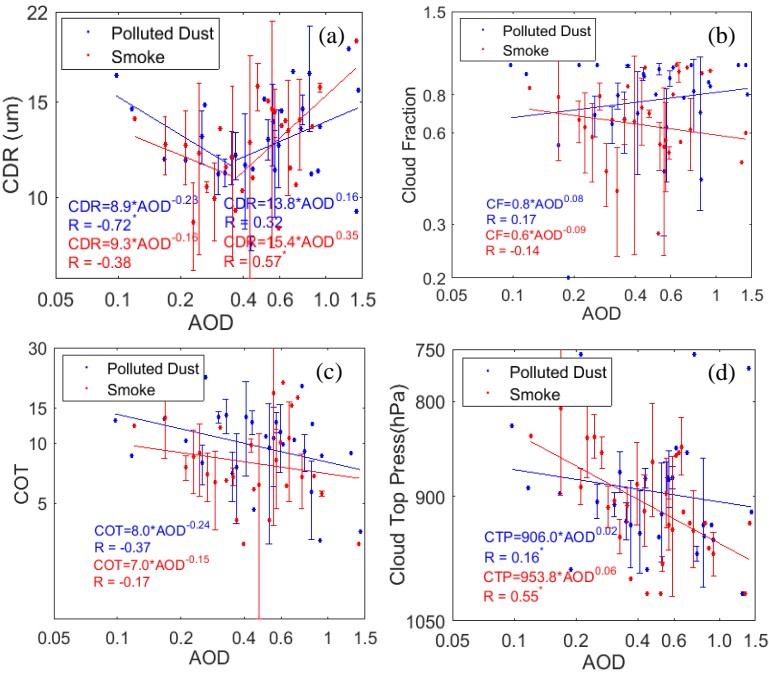

Figure 6. Scatterplots and least square fits of cloud parameters versus AOD over YRD on log-log scale for cases of mixed dust aerosol-cloud layers (blue) and mixed smoke aerosols-cloud layers (red), (a) CDR versus AOD, (b) CF versus AOD, (c) COT versus AOD and (d) CTP versus AOD. The error bars indicate the statistical uncertainties as in Fig. 4.

Eastern China is a region with high concentrations of sulfate, dust and carbonaceous aerosols. In heavily polluted areas, dust aerosols become coated with hygroscopic material, making them effective CCN (Levin et al., 1996; Satheesh et al., 2006). Especially, there are high emissions of smoke by straw burning in summertime. Aerosol - cloud interaction is strongly dependent on the aerosol types, their size distribution and the vertical variation of these, as well as ambient environmental conditions (Patra et al., 2005; Matsui et al., 2006; Dusek et al., 2008; Yuan et al., 2008). Thus, aerosol species are indicative of causal microphysical and radiative effects. Different aerosol types may reveal different patterns of ACI. Here, polluted dust (accounting for 38%) and smoke aerosol (accounting for 40%), which are the two main aerosol types occurring in the YRD, are chosen to investigate the variation of cloud parameters with AOD. Smoke and polluted dust aerosol are discriminated using the CALIOP classification. In addition, they have different efficiency for the absorption of sunlight.





Figure 6 shows the variation of cloud parameters with AOD over the YRD, where data points for mixed polluted dust-warm clouds and mixed smoke aerosols-warm clouds are separated. Figure 6(a) shows that for AOD > 0.3, the slope of the linear regression of CDR against AOD in log-log scale in the presence of smoke is larger than that in the presence of polluted dust, indicating that the ACI is stronger for smoke than for polluted dust. Meanwhile, the cloud fraction is smaller in the presence of smoke, as shown in Figure 6(b). This can be due to the greater efficiency of smoke aerosol particles for the absorption of sunlight than that of dust, resulting in local warming in the presence of smoke aerosol which in turn leads to evaporation of water and thus an increase in small droplets or even complete evaporation of cloud droplets and thus a reduction of cloud cover. Figure 6(c) shows that the cloud optical thickness decreases with an increasing AOD for both aerosol types albeit with a low significance as indicated by the small correlation coefficient R. The cloud optical thickness is, in general, smaller in the presence of smoke aerosol than in the presence of dust. In addition to those mentioned, one thing that probably also contributes to the observed difference between smoke and polluted dust effect is that dust does not absorb sunlight at $0.86\,\mu m$ (Kaufman et al., 2005). Figure 6(d) shows that the slope of linear regression of cloud top pressure against AOD is much stronger for smoke aerosol than that for polluted aerosol, with a correlation coefficient equal to 0.55. Both these two results may be due to the higher absorption efficiency of smoke (Small et al., 2011).





### 3.3.3 Influence of relative humidity on ACI

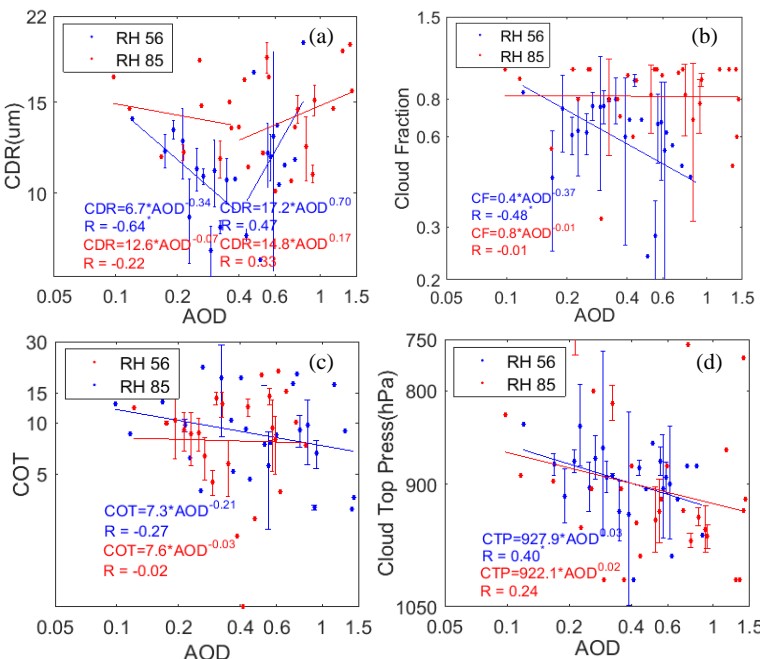

Figure 7. Scatterplots and least square fits of cloud parameters versus AOD over YRD on log-log scale for cases of mixed aerosol-cloud layers under low RH (52%) condition (blue) and mixed aerosol-cloud layers under high RH (83%) condition (red), (a) CDR versus AOD, (b) CF versus AOD, (c) COT versus AOD and (d) CTP versus AOD. The error bars indicate the statistical uncertainties as in Fig. 4.

Relative humidity (RH) is one of the main factors affecting aerosol particle size and cloud formation. A high RH at cloud base has been reported to be associated with how aerosol particles affect cloud properties (Small et al., 2011). Thus, effects of RH need to be accounted for in estimating the relation between aerosol particles and cloud properties, as reported in the literature (Jeong et al., 2007; Loeb and Manalo-Smith, 2005; Quaas et al., 2010).

The cloud properties versus AOD relationships are stratified by three equally sized subsets by RH (at 950hPa), and the mean RH values for each subset are calculated. In figure 7 we show cloud properties as function of AOD for only the lowest RH (56%), representing dry conditions, and for the highest RH (85%, above the deliquescence point of ambient particles). Figure 7(a) shows that the CDR is larger in high relative humidity conditions than in low relative humidity conditions, irrespective of the AOD. It is likely that hygroscopic aerosols grow in size caused by condensation of water vapour (Hanel, 1976; Feingold et al., 2003). The increasing RH further increases the possibility in promoting the cloud droplet





activation and growth of existing cloud as well (Jones et al., 2009). This indicates that high relative

humidity conditions can help the formation of larger cloud droplets due to a higher water vapour content

in the atmosphere. The cloud fraction is much larger in high relative humidity conditions than in low

relative humidity conditions, as shown in Figure 7(b). The impact is more pronounced at high aerosol

load than at low aerosol load. The sensitivity of COT to AOD is weak in both RH conditions for the

whole AOD dataset (Figure 7(c)), indicating that the RH effect in this case is not significant. Figure 7(d)

also shows that there is no significant effect of RH on the relationship between CTP and AOD throughout

the range of AOD. This result may indicate that RH has a minor effect on the relationship between CTP

and AOD.

**3.3.4 Influence of BL thermodynamics and dynamics on ACI**

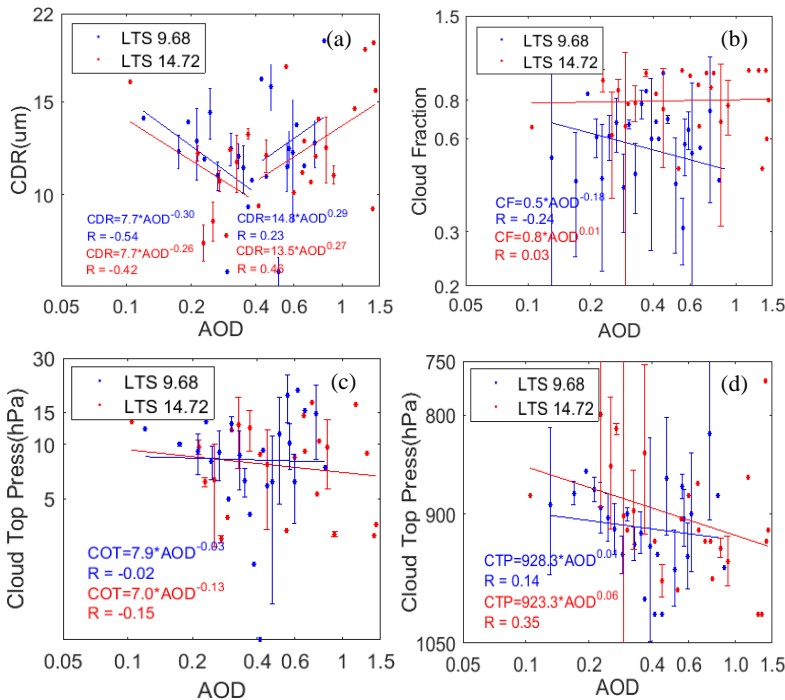

Figure 8. Scatterplots and least square fits of cloud parameters versus AOD over YRD on log-log scale for cases of
mixed aerosol-cloud layers under low LTS condition (blue) and mixed aerosol-cloud layers under high LTS
condition (red), (a) CDR versus AOD, (b) CF versus AOD, (c) COT versus AOD and (d) CTP versus AOD. The
error bars indicate the statistical uncertainties as in Fig. 4.





The LTS is an indicator for the mixing state of the atmospheric layer adjacent to the surface. It describes to some extent the atmosphere's tendency to promote or suppress vertical motion (Medeiros and Stevens, 2011), which in turn affects cloud properties (Klein and Hartmann, 1993).

Figure 8 shows cloud properties as function of AOD for two different LTS conditions: low LTS, with a
mean value equal to 9.68 representing an unstable atmosphere; and high LTS, with a mean value equal to 14.72 representing a stable atmosphere. Figure 8(a) shows that the CDR is larger in unstable atmospheric conditions than in stable conditions, irrespective of the AOD. This indicates that in unstable atmospheric conditions the cloud droplets are larger, which may be due to stronger interaction between aerosols and clouds as a result of better vertical mixing of water vapour. In contrast, the cloud fraction is larger in
stable atmospheric conditions than in unstable conditions for all values of AOD, as shown in Figure 8(b). This demonstrates that stable atmospheric conditions can promote the formation of a cloud (Small, et al., 2011). A high LTS indicates a strong inversion, which prevents vertical mixing and cloud vertical extent, maintaining a well-mixed and moist boundary layer and providing an environment which favours the development of a low cloud cover (Costantino and Breon, et al., 2013). Figure 8(c) and Figure 8(d) show
that there is no significant effect of LTS on the relationships between COT and AOD, and CTP and AOD. This result indicates that other factors may play a large role in determining these relationships.

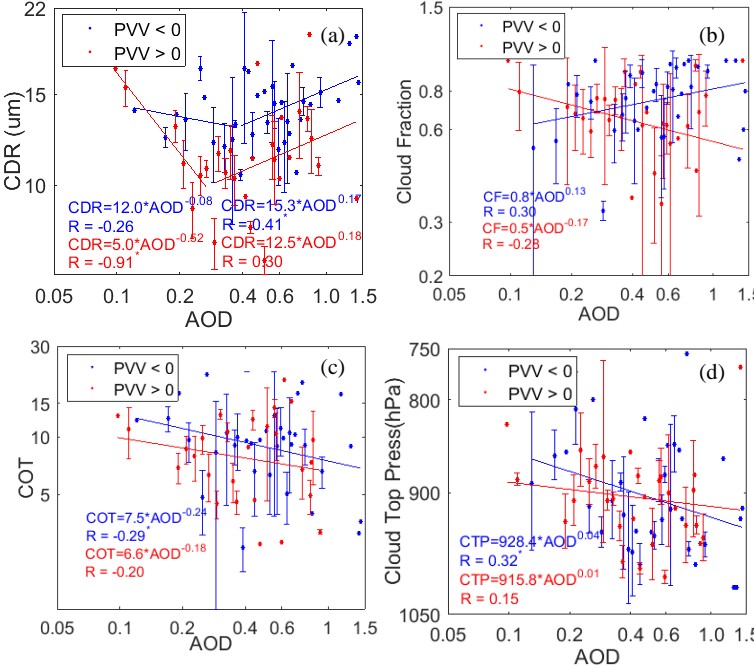





Figure 9. Scatterplots and least square fits of cloud parameters versus AOD over YRD on log-log scale for cases of mixed aerosol-cloud layers under PVV<0 condition (blue) and mixed aerosol-cloud layers under high PVV>0 condition (red), (a) CDR versus AOD, (b) CF versus AOD, (c) COT versus AOD and (d) CTP versus AOD. The error bars indicate the statistical uncertainties as in Fig. 4.

The PVV, a measure of dynamic convection strength, is very important for cloud formation. In particular, the vertical velocity can be used to determine whether a certain region may be susceptible to cloud development or not. That is, the presence of upward motion, as indicated by negative PVV, can enhance ACI as it makes the ambient environment favourable for cloud formation, and vice versa (Jones et al., 2009).

Figure 9(a) shows that the CDR is overall larger in the presence of upward motion of air parcels than for downward motion, irrespective of the aerosol loading. This observation indicates that the upward motion of air parcels can promote the formation of larger cloud droplets, thus enhancing ACI. Figure 9(b) shows that the cloud fraction has a negative correlation with AOD in the presence of downward motion of air parcels but positive correlation with AOD for upward motion, albeit both cases with a low significance.

Finally, the cloud fraction is larger in the presence of upward motion of air parcels than for downward motion of air parcels when AOD is greater than 0.3. This indicates that the upward motion of air parcels can favour cloud development and increase cloud cover in heavily polluted conditions. These results emphasize the importance of vertical velocity when estimating the potential aerosol effect on cloud droplet effective radius and cloud fraction. Figure 9(c) and Figure 9(d) show that there is a weak effect of

pressure vertical velocity on the relationships between COT and AOD, and CTP and AOD. In other words, there is a more significant ACI and a minor meteorological effect.

## 4. Conclusions

Eastern China has been reported to have a high level of anthropogenic emissions, being thereby a hotspot area for studying how cloud microphysical properties are affected by anthropogenic aerosols (Ding et al.,

2013). Based on the near-simultaneous aerosol and cloud retrievals derived from MODIS, CALIOP and CloudSat, together with the ERA Interim Reanalysis data, we investigate the effect of aerosols, where AOD is used as a proxy for aerosol loading, on micro-physical and macro-physical cloud properties over the Yangtze River Delta in summertime for the years 2007 to 2010. In terms of the relative heights of aerosol and cloud layers, well-mixed and separated clouds are defined. Statistical analysis is used to

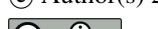



examine the aerosol effects on cloud properties for these two cases. Besides the aerosol impact on CDR, CF, COT and CTP, the influence of environmental conditions, such as RH, LTS, PVV, on the relation between cloud properties and AOD is also studied. In addition, the impact of two different aerosol types, dust and smoke, is explored.

We analyse the COT-CDR and CWP-CDR relationships for well-mixed clouds, showing that both COT and CWP increase with CDR in moderately polluted environments (as defined by AOD<0.3) but decrease with CDR for polluted and heavily polluted cases. Our results indicate that the amount of aerosol can affect the COT-CDR and CWP-CDR relationships. Statistical analysis of the relation between CWP and COT show an increase in CWP with an increasing COT, which is in good agreement

with the findings reported by Costantino and Bréon (2013).

Consistent with previous findings, we find that the CDR initially decreases with increasing AOD, followed by an increase when AOD increases over a value of 0.3. This result is consistent with Twomey's hypothesis that increasing aerosol abundance leads to more numerous but smaller cloud droplets at given constant cloud water content. The positive relation between CDR and AOD may be

caused by microphysical processes, which is coupled with intense vapour competition and evaporation of smaller droplets as a result of a high abundance of aerosol particles. Also, the variation of CF with increasing AOD is analysed and find that CF is little dependent on the AOD for mixed clouds, but shows a positive relation with AOD for well-separated clouds. COT is found to decrease with an increasing aerosol concentration. We argue that the radiative effect and retrieval artefact due to absorbing aerosol

might be important factors in determining this relationship. This effect can result in increased cloud evaporation and reduced cloud cover. Meanwhile, CTP tends to increase as aerosol abundance increases, indicating that the aerosol is prone to expand the horizontal extension.

Furthermore, joint correlative analysis of different aerosols and cloud properties reveal that different aerosol types can have a different effect on aerosol-cloud interaction. The interaction between CDR and

AOD is stronger for smoke aerosols than for dust aerosols in high aerosol loading conditions. At the same time, cloud fraction and COT are smaller for the warm clouds mixed with smoke. Compared to polluted dust aerosol, the relation between cloud top pressure and AOD is much stronger. These results may be caused by a stronger absorption of solar radiation by smoke aerosols than by polluted dust. Therefore, we can conclude that absorbing aerosol plays an important role in the aerosol cloud interaction.



Constrained by relative humidity and boundary thermodynamic and dynamic conditions, the variation of cloud properties in response to aerosol abundance is analysed. In general, high relative humidity can help the formation of larger cloud droplet particles and expand cloud fraction. In contrast, CDR tends to be larger in an unstable atmospheric environment, whereas cloud fraction is larger in a stable atmospheric environment. Dynamically, CDR is larger under upward motion of air parcels, which can be partially attributed to the fact that upward motion of air parcels can promote cloud development. These impacts were pronounced in heavily polluted conditions than in moderately conditions. With regard to COT and CTP, we find that meteorological conditions play a weak role in the interaction between aerosol and cloud. Besides the meteorological controls mentioned above, other factors may be important in generating the relations between aerosol and cloud properties, such as temperature advection. These results suggest that effects of ambient meteorological environments need to be considered when exploring the aerosol indirect effect.

### Acknowledgements

This work was supported by the National Key Research and Development Program of China (No.2016YFD0300101), the 1-3-5 Innovation Project of RADI_CAS (No. Y3ZZ15101A), the National Natural Science Foundation of China (No. 31571565), Open Fund of Key Lab. of ULRMS, MLR (No.KF-2016-02-026) and FCoE, Academy Professorhip. We are grateful to the ease access to MODIS, CALIPSO and CloudSat, provided by NASA and CNES. We also thank ECMWF for providing daily ERA Interim Reanalysis data in our work.

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
