# Peer review of "Satellite observed indications of aerosol effects on warm cloud properties over Yangtze River Delta of China"

_Atmospheric Chemistry and Physics, 2016_

## Referee Comment (RC1) · Anonymous Referee #1 · 1 Jan 2017

The authors analyzed a set of satellite retrievals on cloud properties and AOD to examine the effect of aerosols on the low warm cloud over the Yangtze River Delta region of China. The relationships between AOD and several cloud properties, i.e., CDR, CF, COT, and CTP, were examined by using least square correlation analysis. Furthermore, the physical interaction of cloud and aerosol, different aerosol types and meteorological conditions were also taken into account in the analysis. In general, the manuscript is well organized and the analysis is to some extent comprehensive. It is recommended that the manuscript can be considered for publication after the following specific comments being addressed.

Specific comments:

[Figure]

1. Page 1, Line 21: AOD<0.3 and AOD>0.3

2. Abstract: it would be much helpful if the authors could highlight the overall significance (or implications) of the present study at the end of the abstract (and also in the conclusion section).

3. Page 2, Line 3: change ", and second," to ". Second,"

4. Page 2, Line 10: and clouds, and the aerosol activation efficiency...

5. Page 2, Line 25: and/or

6. Page 3, Lines 19-21: "when using MODIS data" is strange. Rephrase this sentence.

7. Page 4, Line 20: Description of the study region

8. Page 5, Figure 1: show in the plot what is the color-coded legend for, AOD?

9. Page 6, Line 7: delete "also".

10. Page 6, Line 8-9, "CloudSat was the first mission to fly the first..." rephrase this sentence.

11. Page 7, Table 1: reformat the table, especially the first column.

12. Page 8, Line 2: g m-2

13. Page 10, Line 19: replace "CTP" by "CTH"

14. Page 12, Line 20: delete "can".

15. Page 12, Lines 21-22: on the disagreement with the previous findings, can the authors comment on the possible reason?

16. Page 15, Section 3.3.2: what chemical compositions do the smoke aerosols identified by the CALIOP retrievals contain (I presume carbonaceous aerosols)? Please elaborate.

17. Page 17, Figure 7: the discrimination between low and high RH conditions in the Figure caption (52% and 83%) is inconsistent with the numbers shown in the plot (56% and 85%). Please clarify.

18. Page 17, Lines 7-9, "associated with how aerosol particles…": rephrase this sentence.

19. Page 18, Line 10: define "BL".

20. Page 18, Figure 8: delete the second "mixed aerosol-cloud layers under" in the Figure caption.

21. Page 20, Figure 9: delete the second "mixed aerosol-cloud layers under" in the Figure caption.

22. Page 20, Conclusions: the conclusion section is too long. It would be better if the authors could concisely summarize the major key findings of this study, other than listing all of the activities and results, in the Conclusion section.

23. This study focuses on the Yangtze River Delta region, but the discussion of results is somewhat general. It doesn't mention what results are unique for the target region. It would be helpful if the authors compare the results in this study with those obtained from other areas in the world, and comment on if any uniqueness of aerosol effects on clouds in the target YRD region.

---

## Referee Comment (RC2) · Anonymous Referee #2 · 2 Jan 2017

**Review of manuscript doi:10.5194/acp-2016-1000**

**Satellite observed indications of aerosol effects on warm cloud properties over Yangtze River Delta of China**

**by Y.Liu et al.**

**General comments:**

This study investigates the relationship between aerosol and meteorological parameters and low warm clouds properties using satellite observations. The authors focus on summertime periods during 2007-2010 over the Yangtze River Delta, a region characterized by a large variability in aerosol amount and composition. The research questions addressed in this work are highly relevant for an improved understanding of mechanisms of aerosol-cloud interaction and ultimately of aerosol indirect effects. The topic discussed is thus relevant to ACP readers. The manuscript is overall well written and the results are clearly presented. However, major aspects of the paper need to be revised as described in the following general and specific comments to be considered suitable for publication.

- Why only years between 2007-2010 are considered? Given the low availability of satellite observations during these years, as it appears for example in Figures 2-3, to raw more robust conclusions would require a larger sample size. An idea could be to analyze data for the whole acquisition period of CALIPSO (i.e. since 2006).
- It would be beneficial to have a figure/table showing satellite retrieval availability over the analyzed domain and in all figures the sample size should be also reported.
- The uncertainty in the analyzed satellite retrievals should be discussed and related to the significance of the relationships identified between AOD/CDR and other cloud properties. Further, more than half of the reported correlation coefficients are either not significant or very low. I don't see a strong evidence of most of the identified relationship between the analyzed variables, based on such a small sample size, considering the uncertainty in the used retrievals and the absence of significant regression parameters.
- The way results are presented could be improved to have a more fluent and connected discussion on aerosol effects on warm clouds properties instead of presenting a description of each figure as a separate paper section. The authors should integrate all findings in a more general framework including a wider discussion on all analyzed properties and how they relate to each other.

**Specific comments:**

- Page 4, line 4-11: what are the spatio-temporal scales of variability of aerosol and cloud properties and how are they represented by the satellite observations you are analyzing?
- Page 4, line 6-7: how did you analyze "the response to the increase in aerosol loading"? Did you look at AOD temporal trends? Or do you only mean you aim at analyzing the sensitivity of cloud properties to different aerosol loading? By extending your analysis to multiple years you could also look at trends in aerosol loading (if present and if enough data are available).
- Page 5, line 16: Why are you using Collection 5.1 instead of 6?

- Page 6, line 20: What is the vertical resolution of CALIOP/CALIPSO aerosol products?
- Page 9, line 5: Why such few data are available in Figure 2 compared to the other figures (i.e. from Figure 4 on)? A correlation coefficient R of 0.08-0.23 correspond to a coefficient of determination $R^2$ of 0.6-5% which indicate that your regression model is able to explain between 0.6 and 5% of the variability in the data. Further these correlations are not significant. These results need to be better interpreted in the manuscript and the robustness of your finding to be discussed. For example it is very hard to justify that "the correlation between these parameters is negative but weak" at line 14, based on the results presented in Figure 2a. Analysis of longer time series of satellite observations may help in strengthening your conclusions. In all figures the sample size should be also reported.
- In all figures: how are the data aggregated in time? Does each dot represent a daily observation?
- Page 9, line 8: you should include a reference describing the pollution classification based on AOD values.
- Page 9, line 11: this sentence needs to be rephrased. It is not clear what it means the "significance of the difference" and what the p-value refers to.
- Page 10: Are your results consistent with the literature? What type of significant relationship was found between COT/CWP and CDR in other studies? Given the lack of strong evidence in your results a wider discussion on what has been found so far in the literature is necessary.
- Page 11, line 8: why in all panels of Figure 4 there are many more points than in Figure 2 even considering only the mixed aerosol-cloud layers?
- Page 12, line 4: Please be more clear in explaining how you infer that "CDR is ~ 3times stronger…"
- Page 12, line 5: the discussion of pollution levels as a function of AOD should be introduced earlier in the paper given it is used since the first analyses presented. You should also discuss why you are choosing a threshold of 0.3 instead of 0.4 and in the cited reference.
- Page 14, Figure 5: Why do you not separate cases with AOD> (<) 0.3 in all panels? At least in panel b there could be a different relationship if this threshold is applied.

**Technical corrections:**

- Page 4, line 21: Figure 1 is not referenced in the manuscript, so it could be added where you introduce the analyzed domain.
- Page 6, line 7: the CALIPSO acronym needs to be defined
- Page 8, line 2: -2 needs to be superscript
- Page 5, line 12: Since you are using only data from MODIS Aqua, the reference to the Terra satellite should be removed everywhere in the paper.
- Page 11, line 12: a space is missing between "and" and "σ"
- Page 11, line 15: remove "a" before "cloud parameters"
- Is there a way to differentiate the figures? Using only red and blue in all figures/panels is misleading since the reader may associate a specific color to a specific property.

---

## Author Response (AR1)

**Reply to comments on "Satellite observed indications of aerosol effects on warm cloud properties over Yangtze River Delta of China"**

March 28, 2017

We thank the reviewer's thoughtful comments which are helpful not only for this manuscript but also for our future research. Our replies for all the comments are shown below.

**Specific comments**
**1. Comments: (1) Page 1, Line 21: AOD<0.3 and AOD>0.3.**
**Answer**: We made this change (see pg.1 line 23).

**2. Comments: (2) Abstract: it would be much helpful if the authors could highlight the overall significance (or implications) of the present study at the end of the abstract (and also in the conclusion section).**
**Answer**: We made this change in the abstract (see pg.1) and in the conclusion section (see pg.25-27).
Page 1, line 35-39: Text was added as:'Overall, the present study provides an understanding of the impact of aerosols on cloud properties over the YRD. In addition to the amount of aerosol particles (or AOD), evidence is provided that aerosol types and ambient environmental conditions need to be considered to understand the observed relationships between cloud properties and AOD.'
Page 27, line 16-17: Text was added as:' In summary, this study will greatly help us to understand the mechanisms of aerosol-cloud interaction and ultimately of aerosol indirect effects over the YRD.'

**3. Comments: (3) Page 2, Line 3: change ", and second," to ". Second,"**
**Answer**: We made this change (see pg.2 line 6).

**4. Comments: (4) Page 2, Line 10: and clouds, and the aerosol activation efficiency…**
**Answer**: We made this change (see pg.2 line 13).

**5. Comments: (5) Page 2, Line 25: and/or**
**Answer**: We made this change (see pg.2 line 28).

**6. Comments: (6) Page 3, Lines 19-21: "when using MODIS data" is strange. Rephrase this sentence.**
**Answer**: We made this change (see pg.3 line 23-25). "Costantino and Breon (2013)

and Jones et al. (2009) found that the aerosol indirect effect is stronger for well-mixed clouds than for well-separated clouds (in well-mixed aerosol and cloud layers are physically interacting, as further explained in Section 2) when using MODIS data." has been changed to "Costantino and Breon (2013) and Jones et al. (2009), using MODIS data, found that the aerosol indirect effect is stronger for well-mixed clouds than for well-separated clouds (in well-mixed aerosol and cloud layers are physically interacting, as further explained in Section 2)." in the revised manuscript.

**7. Comments: (7) Page 4, Line 20: Description of the study region**
**Answer**: We made this change (see pg.4 line 25).

**8. Comments: (8) Page 5, Figure 1: show in the plot what is the color-coded legend for, AOD?**
**Answer**: Yes, it represents for AOD. We improved the color-coded legend as shown in Figure 1 in the revised manuscript (see pg.5).

**9. Comments: (9) Page 6, Line 7: delete "also".**
**Answer**: We made this change (see pg.6 line 17).

**10. Comments: (10) Page 6, Line 8-9, "CloudSat was the first mission to fly the first…" rephrase this sentence.**
**Answer**:"CloudSat was the first mission to fly the first…" has been changed to "CloudSat carries the CPR (Cloud Profiling Radar), i.e. the first satellite-based millimeter-wavelength cloud radar to detect the vertical information on different sized cloud droplets" in the revised manuscript (see pg.6 line 13-15).

**11. Comments: (11) Page 7, Table 1: reformat the table, especially the first column.**
**Answer**: We made this change (see pg.7).

**12. Comments: (12) Page 8, Line 2: g m-2**
**Answer**: We made this change (see pg.8 line 7).

**13. Comments: (13) Page 10, Line 19: replace "CTP" by "CTH"**
**Answer**: We made this change (see pg.13 line 9).

**14. Comments: (14) Page 12, Line 20: delete "can".**
**Answer**: We made this change (see pg.15 line 16).

**15. Comments: (15) Page 12, Lines 21-22: on the disagreement with the previous findings, can the authors comment on the possible reason?**
**Answer**: Yes, we reanalyzed more data for the considered years and found a clearer relationship between cloud fraction and AOD. Meanwhile, we gave possible reasons for this finding that are not in the disagreement with the previous findings (see pg.15 lines 19-25).

**16. Comments: (16) Page 15, Section 3.3.2: what chemical compositions do the smoke aerosols identified by the CALIOP retrievals contain (I presume carbonaceous aerosols)? Please elaborate.**

**Answer**: The CALIOP classification of aerosol subtypes utilizes a depolarization ratio, integrated attenuated backscatter coefficient, surface type and information on whether the layer is elevated or not. The CALIOP algorithm is based on physical properties and does not distinguish aerosol by chemical composition. However, according to previous studies, the smoke particles are observed to strongly absorb solar radiation, particularly at ultraviolet (UV) and visible (VIS) wavelengths. Hence, we can estimate that the aerosols we studied contain carbonaceous aerosols (Mielonen et al., 2009; Ford et al., 2013; Nowottnick et al., 2015) (see pg.19 line 2).

**17. Comments: (17) Page 17, Figure 7: the discrimination between low and high RH conditions in the Figure caption (52% and 83%) is inconsistent with the numbers shown in the plot (56%and 85%). Please clarify.**

**Answer**: Now as the analyzed dataset was different, also the result changed. We rephrased the sentence in the revised manuscript (see pg. 20-21). Here, we show cloud properties as function of AOD for only the lowest RH (31%), representing dry conditions, and the highest RH (91%, above the deliquescence point of ambient particles). We made this change (see pg. 20-21).

**18. Comments: (18) Page 17, Lines 7-9, "associated with how aerosol particles…": rephrase this sentence.**

**Answer**: "reported to be associated with how aerosol particles…" has changed to" reported to affect the relation between aerosol particles and cloud properties…" in the revised manuscript (see pg.20 lines 14-15).

**19. Comments: (19) Page 18, Line 10: define "BL".**

**Answer**: "BL" has been changed to "boundary layer" in the revised manuscript (see pg.22 line 1).

**20. Comments: (20) Page 18, Figure 8: delete the second "mixed aerosol-cloud layers under" in the Figure caption.**

**Answer**: We made this change (see pg.20 line 4, see pg.22 line 4, see pg.23 line 15).

**21. Comments: (21) Page 20, Figure 9: delete the second "mixed aerosol-cloud layers under" in the Figure caption.**

**Answer**: We made this change (see pg.22 line 4).

**22. Comments: (22) Page 20, Conclusions: the conclusion section is too long. It would be better if the authors could concisely summarize the major key findings of this study, other than listing all of the activities and results, in the Conclusion section.**

**Answer**: Yes. We reorganized the conclusion and summarized the major key findings of this study concisely in the conclusion section (see pg.25-27). Here, we present the conclusion section below.

**Conclusion**

The high level of anthropogenic emissions in Eastern China render this area an important hotspot for studying how cloud microphysical properties are affected by anthropogenic aerosols (Ding et al., 2013). Based on the near-simultaneous aerosol and cloud retrievals provided by MODIS, CALIOP and CloudSat, together with the ERA Interim Reanalysis data, we investigated the effect of aerosols, with AOD used as a proxy for the aerosol loading, on micro-physical and macro-physical cloud properties over the Yangtze River Delta for the years 2007 to 2010. In terms of the relative heights of aerosol and cloud layers, well-mixed and separated clouds were defined. A statistical analysis was used to examine the aerosol effects on cloud properties for these two cases. Besides the aerosol impact on CDR, CF, COT and CTP, also the influence of environmental conditions, such as RH, LTS and PVV, on the relation between cloud properties and AOD was studied. In addition, the impact of two different aerosol types, dust and smoke, was explored.

The analysis of the COT-CDR and CWP-CDR relationships for well-mixed clouds indicated that they are affected by the aerosol loading. A statistical analysis of the relation between CWP and COT showed an increase in CWP with an increasing COT, which is in a good agreement with the findings reported by Costantino and Bréon (2013).

Consistent with previous findings, we found that the CDR initially decreases with increasing AOD, followed by an increase after AOD reaches a value of 0.35. This result is consistent with Twomey's hypothesis that increasing aerosol abundance leads to more numerous but smaller cloud droplets at given constant cloud water content. The positive relation between CDR and AOD may be caused by microphysical processes, which is coupled with intense vapour competition and evaporation of smaller droplets as a result of a high abundance of aerosol particles. Also, the analysis of the variation of CF with increasing AOD showed that CF varies with AOD in a way similar to that of CDR. This finding differs from those by Koren et al. (2008) and Small et al. (2011) who observed that an increase in the cloud cover with an increasing AOD, followed by a decrease with higher AOD. COT was found to decrease with an increasing AOD. We argue that the radiative effect and retrieval artefact due to absorbing aerosol might be important factors in determining this relationship. This effect can result in increased cloud evaporation and reduced cloud cover. Meanwhile, CTP tends to increase as aerosol abundance increases, indicating that the aerosol is prone to expand the horizontal extension. In other words, we found that for well-mixed clouds over the YRD, the CDR becomes smaller with the increase of AOD in moderately polluted conditions which in principles in line with the Twomey effect, yet, the cloud fraction indicates a weak decrease which could be attributed only to the weak influence of evaporation caused by absorption of aerosols.

On the other hand, in polluted and heavily polluted conditions, a reduced cloud coverage can result in more solar radiation reaching the surface, causing surface

heating and thus raises the surface temperature, which then destabilizes the atmosphere. The resulting advection transports water vapour from the surface to higher levels in the atmosphere, therefore producing more cloud. Meanwhile, CDR becomes larger as a result of the stronger water vapour competition in polluted and heavily polluted conditions. The COT decreases with the increasing values of AOD throughout the AOD range due to the radiative effect and possible retrieval artefacts. The behaviour of CTP is consistent with that of COT, with the cloud getting thinner but with larger cover, so that CTP becomes larger with an increasing AOD.

Furthermore, joint correlative analysis of different aerosol and cloud properties revealed that smoke aerosols have a stronger impact on aerosol-cloud interaction due to their stronger absorption of solar radiation compared with polluted dust. Therefore, we can conclude that absorbing aerosols play an important role in the aerosol cloud interaction.

Constrained by relative humidity and boundary thermodynamic and dynamic conditions, the variation of cloud properties in response to aerosol abundance was analysed. In general, a high relative humidity can promote the formation of larger cloud droplets and expand cloud formation, irrespective of the vertical or horizontal level. With regard to LTS, stable atmospheric conditions can enhance the cloud cover horizontally. However, unstable atmospheric conditions can be helpful for the formation of thicker and higher clouds. Dynamically, an upward motion of air parcels can also facilitate the formation of thicker and higher clouds. Besides the meteorological controls mentioned above, other factors may be important in generating relations between aerosol and cloud properties, such as temperature advection. These results suggest that effects of ambient meteorological environments need to be considered when exploring the aerosol indirect effect. In summary, this study will greatly help us to understand the mechanisms of aerosol-cloud interaction and ultimately of aerosol indirect effects over the YRD.

**23. Comments: (23) This study focuses on the Yangtze River Delta region, but the discussion of results is somewhat general. It doesn't mention what results are unique for the target region. It would be helpful if the authors compare the results in this study with those obtained from other areas in the world, and comment on if any uniqueness of aerosol effects on clouds in the target YRD region.**

**Answer**: Yes. We added some key sentences into the manuscript to mention the unique result for the YRD. For example, the sentence "This outcome is not in agreement with the findings of Koren et al. (2008) and Small et al. (2011)." in the manuscript (see pg.15 line 19) and the sentence 'This study shows that the COT-CDR and CWP-CDR relationships are not unique, but affected by an atmospheric aerosol loading.' in the abstract (see pg.1 line 20-21).

References

Mielonen, T., Arola, A., Komppula, M., Kukkonen, J., Koskinen, J., de Leeuw, G., and Lehtinen, K. E. J.: Comparison of CALIOP level 2 aerosol subtypes to aerosol types

derived from AERONET inversion data, Geophys. Res. Lett., 36, L18804, doi:10.1029/2009GL039609, 2009.

Ford, B., and Heald, C. L.: Aerosol loading in the Southeastern United States: reconciling surface and satellite observations, Atmos. Chem. Phys., 13, 9269-9283, doi:10.5194/acp-13-9269-2013, 2013.

Nowottnick, E.P., Colarco, P.R., Welton, E.J., and Silva, A. da.: Use of the CALIOP vertical feature mask for evaluating global aerosol models, Atmos. Meas. Tech., 8, 3647-3669, doi:10.5194/amt-8-3647-2015, 2015.

Koren, I., Martins, J.V., Remer, L.A., Afargan, H.: Smoke invigoration versus inhabitation of clouds over the Amazon, Science, 321, 946-949, doi: 10.1126/science. 1159185, 2008.

Small, J. D., Jiang, J.H., Su, H., and Zhai, C.: Relationship between aerosol and cloud fraction over Australia, Geophys. Res. Lett., 38, L23802, doi:10.1029/2011GL049404, 2011.

**Reply to comments on "Satellite observed indications of aerosol effects on warm cloud properties over Yangtze River Delta of China"**

March 28, 2017

We would like to appreciate the reviewer for the detailed and valuable comments which helped us a lot to improve the manuscript. Our replies to all the comments are shown below.

**General comments**

**1. Comments: (1) Why only years between 2007-2010 are considered? Given the low availability of satellite observations during these years, as it appears for example in Figures 2-3, to raw more robust conclusions would require a larger sample size. An idea could be to analyze data for the whole acquisition period of CALIPSO (i.e. since 2006).**

**Answer**: This is correct, since the data covering the study area are only available from January 2007 to December 2010. We reanalyze all the data for the whole acquisition period between 2007 and 2010, rather than just summertime data. This issue is shown throughout the revised manuscript (all the figures were changed/modified in this respect).

**2. Comments: (2) It would be beneficial to have a figure/table showing satellite retrieval availability over the analyzed domain and in all figures the sample size should be also reported.**

**Answer**: Yes, we totally agree. We added the spatial and time series analysis of aerosol and cloud parameters over the analyzed domain in the revised manuscript, as shown in Figure 1-2 and Table 1 (see pg.9-10 in the revised manuscript). Also, we reported the sample size in all figures in the revised manuscript.

[Figure]

Figure 1. Spatial distributions of AOD (a), CDR (b), CF (c), COT (d), CWP (e) and CTP (f) averaged over all years between 2007 and 2010.

[Figure]

Figure 2. Time series of the monthly averaged values of AOD (a), CDR (b), CF (c), COT (d), CWP (e) and CTP (f) for all months between 2007 and 2010. Month 1 is January.

Table 1. The sample sizes of all months for each parameter

| Parameters | Jan | Feb | Mar | Apr | May | Jun | Jul | Aug | Sep | Oct | Nov | Dec | Total |
|---|---|---|---|---|---|---|---|---|---|---|---|---|---|
| AOD | 5428 | 3332 | 3892 | 4704 | 5598 | 3638 | 5944 | 6630 | 4306 | 6728 | 6110 | 6400 | 62710 |
| CDR | 794 | 669 | 365 | 679 | 714 | 872 | 1228 | 2013 | 1514 | 1281 | 895 | 582 | 11606 |
| COT | 886 | 747 | 392 | 732 | 748 | 915 | 1298 | 2072 | 1539 | 1329 | 967 | 627 | 12232 |
| CWP | 1226 | 1125 | 620 | 1310 | 1226 | 1245 | 1490 | 2187 | 1929 | 1715 | 1261 | 867 | 16201 |
| CF | 1398 | 994 | 537 | 955 | 993 | 1065 | 1671 | 2650 | 1996 | 1811 | 1373 | 1119 | 16562 |
| CTP | 1398 | 994 | 537 | 955 | 993 | 1065 | 1671 | 2650 | 1996 | 1811 | 1373 | 1119 | 16562 |

**3. Comments: (3) The uncertainty in the analyzed satellite retrievals should be discussed and related to the significance of the relationships identified between AOD/CDR and other cloud properties. Further, more than half of the reported correlation coefficients are either not significant or very low. I don't see a strong evidence of most of the identified relationship between the analyzed variables, based on such a small sample size, considering the uncertainty in the used retrievals and the absence of significant regression parameters.**

**Answer**: According to these comments, we added a subsection (section 3.4 Error sources and uncertainties) into the results and discussions section (see pg.24-25). It describes uncertainties in the satellite retrievals and the significance of the relationships identified between AOD/CDR and other cloud properties. The relationships between the analyzed variables became more robust with a larger sample size, as shown in the revised manuscript. Here, we present the section 3.4 Error sources and uncertainties below.

**3.4 Error sources and uncertainties**

Caution is warranted in investigating the satellite-derived relations between aerosol and cloud properties. Uncertainties in satellite data may results from assumptions on the aerosol size distribution used in the retrieval process, imperfect cloud detection resulting in residual clouds leading to high AOD values, effects of relative humidity on aerosol parameters, and dynamic effects (Yuan et al., 2008). Below we discuss several potential factors that may have affected the interaction between aerosols and clouds in our analysis.

Firstly, the correlation between AOD and cloud parameters may be influenced by aerosol size distributions (Small et al., 2011). Since the MODIS retrieval does not provide aerosol size information, it is better to explore the seasonal differences in the observed ACI due to the difference in aerosol emissions between the different seasons. However, the relatively low number of MODIS-CALIPSO coincidences limits the further binning of the data required to investigate this issue. Secondly, what it comes to the occurrence of cloud contamination in the AOD dataset, this is a universal and one of the most difficult problems in aerosol retrieval. Cloud detection is usually not perfect, so that undetected, or residual, clouds contaminate the retrieval area which leads to AOD overestimation and in turn affects the relation between aerosol and cloud properties (e.g. Sogacheva et al., 2017). A study by Mei et al. (2016), comparing their MERIS cloud mask with two independent data sets, shows that on the order of 70-90% of the cases are correctly classified as cloud free. This result is in good agreement with that from a dedicated study on a consistency between aerosol and cloud retrievals from the same instrument which showed that about 20% of the pixels may be mis-classified (Klueser, 2014). In this study, the samples with AOD values greater than 1.5 were excluded as a rough attempt to exclude cloud-contaminated AOD to reduce the uncertainty in the observed ACI. Thirdly, Feingold et al. (2003) reported that water vapour swelling increases the AOD. Sheridan et al. (2001) showed an important role of hygroscopic growth in determining the AOD for sea salt aerosols. The effect of humidity on the ACI has been discussed

in Section 3.3.3. Finally, Young (1993) reported that ACI is influenced by dynamics through modifying radiative and thermodynamic heating. Jones et al. (2009) emphasized the importance of vertical mixing velocity in cloud formation and ACI as discussed in Sections 3.3.4 and 3.3.5. As reported by Yuan et al. (2010), the potential artefacts above mentioned do not seem to be the primary cause for the observed relationship between aerosol and cloud parameters. Further investigations are needed to fully analyse and explain the observed phenomena.

**4. Comments: (4) The way results are presented could be improved to have a more fluent and connected discussion on aerosol effects on warm clouds properties instead of presenting a description of each figure as a separate paper section. The authors should integrate all findings in a more general framework including a wider discussion on all analyzed properties and how they relate to each other.**

**Answer**: Yes. We integrated all our findings into a more general framework, including a wider discussion on all the analyzed properties and how they are related to each other. For example, we added the sentence "Prior to investigating the aerosol impact on warm cloud properties, a general analysis of cloud properties and the effect of aerosol loading on the relations between them are discussed below." into section 3.1.2 (see pg.11 line 10-11).

The sentence "In this section we examine the responses of various cloud properties to the increasing AOD for well-separated and well-mixed clouds, respectively." was added into section 3.2 (see pg.14 lines 9-10).

The text "Based on the above findings, we conclude that for well-mixed clouds in the YRD, the CDR shows a decrease with an increasing AOD under moderately-polluted conditions, followed by an increase under polluted and heavily-polluted conditions due to the intense water vapour competition. The cloud cover behaves qualitatively similar to CDR in response to changing values of AOD. Meanwhile, cloud optical depth becomes smaller and cloud top pressure becomes larger with increasing AOD over the whole range of AOD values." was added into section 3.2 (see pg.16 lines 22-27).

The text "Feingold et al. (2001) reported that the aerosol indirect effect depends highly on the aerosol hygroscopicity and pressure vertical velocity. Wang et al. (2014) demonstrated that the observed interaction between aerosol and cloud can be affected by the dynamical and thermodynamical processes in cloud systems. Therefore, to explore the meteorological impact on the interaction between aerosol and cloud observed over the YRD, we classify the data for various meteorological parameters, including RH (this section), LTS and PVV (Section 3.3.4)." was added into section 3.3.3 (see pg.20 lines 7-12).

We also reorganized the sentences in the conclusion section (see pg.25-27).

**Specific comments**
**1. Comments: (1) Page 4, line 4-11: what are the spatio-temporal scales of variability of aerosol and cloud properties and how are they represented by the**

**satellite observations you are analyzing?**

**Answer**: The spatial and temporal variability in aerosol and cloud properties are shown in Figures 2 and 3 in the revised manuscript, respectively. We can see a decreasing north-south pattern in AOD in Figure 2a, with the highest values found in the northeast area. CDR behaves similar to AOD, except that the highest values are found in the northernmost area. Contrary to AOD, both COT and CWP show an increasing north-south pattern. Furthermore, the spatial distributions of COT and CWP are remarkably similar to each other.

The monthly-averaged values of AOD and CDR were highest in June, while December showed the lowest monthly-average value for AOD. Overall, the temporal variabilities of COT and CWP were similar, with lowest monthly averages in summer and highest averages in winter. The temporal patterns of CF and CTP were similar throughout the year (see pg.9-10).

**2. Comments: (2) Page 4, line 6-7: how did you analyze "the response to the increase in aerosol loading"? Did you look at AOD temporal trends? Or do you only mean you aim at analyzing the sensitivity of cloud properties to different aerosol loading? By extending your analysis to multiple years you could also look at trends in aerosol loading (if present and if enough data are available).**

**Answer**: We analyzed "the response to the increase in aerosol loading" in two different ways. The first one was to look at AOD temporal trends (as shown in figure 3 in the revised manuscript) and the second one was to analyze the sensitivity of cloud properties to different aerosol loading (this issue is shown through the whole manuscript). We added a spatial and time series analysis of aerosol and cloud parameters over the analyzed domain, as shown in Figure 2 and Figure 3 in the revised manuscript (see pg.9-11).

**3. Comments: (3) Page 5, line 16: Why are you using Collection 5.1 instead of 6?**

**Answer**: The MODIS Collection 05 Level 2 daily product provides cloud and aerosol properties at 10 km×10 km and 1 km×1 km (5 km×5 km) spatial resolution, respectively. The reason why we chose the MODIS Collection 5.1 is the following: most of previous researches were based on MODIS Collection 5.1 and it is easy to compare the results in our study with others' using the same data. Also, Collection 6 (C6) Aqua L2 production began in Dec 2013.

**4. Comments: (4) Page 6, line 20: What is the vertical resolution of CALIOP/CALIPSO aerosol products?**

**Answer**: The vertical resolution of the CALIOP layer product varies with altitudes: 30 m for h =0 - 8.2 km, 60 m for h = 8.2 -20.2 km, and 180 m for h = 20.2 - 30.1 km, whereas the horizontal resolution is 5 km (Liu et al., 2009). This was added into the revised manuscript (see pg.6 lines 24-26).

**5. Comments: (5) Page 9, line 5: Why such few data are available in Figure 2 compared to the other figures (i.e. from Figure 4 on)? A correlation coefficient R of**

**0.08-0.23correspond to a coefficient of determination R² of 0.6-5% which indicate that your regression model is able to explain between 0.6 and 5% of the variability in the data. Further these correlations are not significant. These results need to be better interpreted in the manuscript and the robustness of your finding to be discussed. For example it is very hard to justify that "the correlation between these parameters is negative but weak" at line 14, based on the results presented in Figure 2a. Analysis of longer time series of satellite observations may help in strengthening your conclusions. In all figures the sample size should be also reported.**

**Answer**: All CDR, COT, CWP, CF and CTP data shown in the figures in the manuscript are averaged over AOD bins, from 0.05 to 1.5 by a step of 0.02 on a log-log scale.

However, not every CALIPSO shot has all the corresponding value for AOD, CDR, COT, CWP, CF or CTP. For example, some shots have AOD and CDR values but not COT values, while some shots have AOD and COT values but not CDR values. This reduces the sample size to some extent when considering the relationship between CDR and COT. We have reported the sample size in all the figures in the revised manuscript.

We have interpreted the results in a better way in the revised manuscript (see pg.12), and the robustness of the findings is now discussed in Section 3.4 (see pg.24-25).

**6. Comments: (6) In all figures: how are the data aggregated in time? Does each dot represent a daily observation?**

**Answer**: The time-coincidence of retrievals was assured using datasets from the same date by the A-train coordinated orbits of Aqua and CALIPSO. The detailed data preprocessing is as follows:

1. In this study, all the tracks (509) of CALIPSO covering the target study areas between 2007 and 2010 were selected. According to the latitude-longitude pairs of the profiles, all the profiles (60311) covering the target area (27°N-34°N and 115°E-122°E)were further extracted.

2. When CALIPSO shot detected the presence of aerosol, we averaged the MODIS aerosol retrievals within a radius of 50 km from the CALIPSO target. Likewise, we averaged the MODIS cloud retrievals within a radius of 5 km from the CALIPSO target. For meteorological properties, we chose the value of the footprint that was nearest to the CALIPSO target. Then, every CALIPSO shot had its corresponding AOD, CDR, COT, CWP, CF and CTP (if the value was available).

3. All CDR, COT, CWP, CF and CTP data shown in the figures in the manuscript were averaged over AOD bins, from 0.05 to 1.5 by a step of 0.02 on a log-log scale.

Therefore, each dot does not represent a daily observation in the figures in the manuscript.

**7. Comments: (7) Page 9, line 8: you should include a reference describing the pollution classification based on AOD values.**

**Answer**: We first explored the response of CDR to the increasing AOD in mixed aerosol-cloud layers and found that CDR decreases with increasing AOD in moderately polluted conditions (AOD < 0.35). In polluted and heavily polluted conditions (AOD > 0.35), however, CDR increases with increasing AOD. Here we discriminate between moderately (AOD < 0.35), polluted (AOD >= 0.35 and AOD <=0.8) and heavily polluted (AOD >0.8) conditions. These limits are somewhat arbitrary, however the AOD of 0.35 is based on analysis presented in section 3.2 where conditions change at about this value (see pg. 12).

**8. Comments: (8) Page 9, line 11: this sentence needs to be rephrased. It is not clear what it means the "significance of the difference" and what the p-value refers to.**

**Answer**: Student's t-test was used to determine whether two sets of data were significantlydifferent from each other. The p-value is defined as the probability of obtaining a result equal to or "more extreme" than what was actually observed, when the null hypothesisis true. We rephrased the sentence in the revised manuscript (see pg.12).

**9. Comments: (9) Page 10: Are your results consistent with the literature? What type of significant relationship was found between COT/CWP and CDR in other studies? Given the lack of strong evidence in your results a wider discussion on what has been found so far in the literature is necessary.**

**Answer**: In this study, we explored the response of CDR and CWP to the increasing value of COT under different pollution conditions. Costantino and Bréon (2013) compared the CDR-COT relationship of mixed and unmixed aerosol-cloud layers and found an increase in CDR with an increasing COT, followed by a decrease with higher COT in both cases (mixed and separated aerosol-cloud layers) (see pg.12 lines 24-27). They also reported that cloud water amount increases with an increasing cloud optical thickness. We added this reference into the revised manuscript (see pg.13 lines 20-21).

**10. Comments: (10)Page 11, line 8: why in all panels of Figure 4 there are many more points than in Figure 2 even considering only the mixed aerosol-cloud layers?**

**Answer**: Please see our answer to comment 5.

**11. Comments: (11) Page 12, line 4: Please be more clear in explaining how you infer that "CDR is ~ 3times stronger…"**

**Answer**: "…the relation between AOD and CDR is ~3 times stronger for the mixed layers than for separated layers…" can be inferred from the different slopes of the lines for mixed (-0.38) and separated (-0.14) layers in the previous manuscript. Now as the analyzed dataset was different, also the result changed. We rephrased the sentence in the revised manuscript (see pg.14-15).

**12. Comments: (12) Page 12, line 5: the discussion of pollution levels as a function of AOD should be introduced earlier in the paper given it is used since the first**

**analyses presented. You should also discuss why you are choosing a threshold of 0.3 instead of 0.4 and in the cited reference.**

**Answer**: Yes, we totally agree. We modified the revised manuscript (see pg.12 lines 8-13). Now we reanalyzed all the data for the whole acquisition period between 2007 and 2010, rather than just summertime data, and we found that a threshold of 0.35 is better for differentiating between the different pollution levels over the YRD. CDR shows a negative relation with AOD in moderate polluted conditions (AOD < 0.35). In polluted and heavily polluted conditions (AOD > 0.35), however, CDR increases with an increasing AOD (see pg.12).

**13. Comments: (13) Page 14, Figure 5: Why do you not separate cases with AOD> (<) 0.3 in all panels? At least in panel b there could be a different relationship if this threshold is applied.**

**Answer**: Yes, according to these comments, we separated a case with AOD> (<) 0.35 in panel b and found a different result (see pg.14-15). However, we did not separate cases with AOD> (<) 0.35 in all the panels, as there was no significant difference in the panels c and d. Note that the threshold of AOD is 0.35 in the revised manuscript.

**Technical comments**
**1. Comments: (1) Page 4, line 21: Figure 1 is not referenced in the manuscript, so it could be added where you introduce the analyzed domain.**

**Answer**: Yes, we added "figure 1" where we introduce the analyzed domain in the revised manuscript (see pg.4 line 26).

**2. Comments: (2) Page 6, line 7: the CALIPSO acronym needs to be defined**

**Answer**: "…CALIPSO and CloudSat are flying in the so-called…" has been changed to "CALIPSO (Cloud-Aerosol Lidar and Infrared Pathfinder Satellite Observations) and CloudSat are flying in the so-called…" in the revised manuscript (see pg.6 line 11).

**3. Comments: (3) Page 8, line 2: -2 needs to be superscript**
**Answer**: We made this change in the revised manuscript (see pg.8 line 7).

**4. Comments: (4) Page 5, line 12: Since you are using only data from MODIS Aqua, the reference to the Terra satellite should be removed everywhere in the paper.**

**Answer**: We made this change. The reference to the Terra satellite has been removed in the revised manuscript. "The MODIS sensors, onboard the Terra and Aqua satellites…" has changed to "The MODIS sensor, onboard the Aqua satellites…" (see pg.5 line 17), and "Along with the Aqua and Terra satellites…" has been changed to "Along with the Aqua satellites…" (see pg.6 line 11).

**5. Comments: (5) Page 11, line 12: a space is missing between "and" and "σ"**
**Answer**: We made this change (see pg.14 line 7).

**6. Comments: (6) Page 11, line 15: remove "a" before "cloud parameters"**

**Answer**: We made this change. "The strength of the interaction between cloud properties and AOD is quantified here as the slope of the line describing the relation between a cloud parameters and AOD, on a log-log scale, as obtained by linear regression." has changed to "The strength of the interaction between cloud properties and AOD is quantified here as the slope of the line describing the relation between cloud parameters and AOD, on a log-log scale, as obtained by linear regression." (see pg.14 line 13).

**7. Comments: (7) Is there a way to differentiate the figures? Using only red and blue in all figures/panels is misleading since the reader may associate a specific color to a specific property.**

**Answer**: Yes, we totally agree. We differentiated the figures using different markers and colors together for different cloud properties. We made this change throughout the manuscript. Here, we just take figure 3 for example below.

[Figure]

Figure 3. Scatterplots of cloud parameters versus AOD over YRD on log-log scale for cases of separated (blue) and mixed (red) aerosol-cloud layers, (a) CDR versus AOD, (b) CF versus AOD, (c) COT versus AOD and (d) CTP versus AOD. The lines present the least squares fits and the resulting relations are presented in each figure. Error bars represent the confidence level of the mean cloud parameters' value for each AOD bin, i.e. the statistical uncertainties, expressed as $\sigma/(n-2)$ , where n is the number of cases within the AOD bin and $\sigma$ is the standard deviation of cloud properties.

**References**

Feingold, G., Remer, L.A., Ramaprasad, J., Kaufman, Y.J., 2001. Analysis of smoke impact on clouds in Brazilian biomass burning regions: an extension of Twomey's approach. J. Geophys. Res. 106 (No.D19), 22907-22922.

Costantino, L. and Bréon, F.M.: Aerosol indirect effect on warm clouds over South-East Atlantic, from co-located MODIS and CALIPSO observations, Atmos. Chem. Phys., 13, 69-88, 2013.

Liu, Z., Vaughan, M., Winker, D., Kittaka, C., Getzewich, B., Kuehn, R., Omar, A., Powell, K., Trepte, C., Hostetler, C., 2009. The CALIPSO lidar cloud and aerosol discrimination: Version 2 algorithm and initial assessment of performance. J. Atmos. Ocean. Technol. 26 (7), 1198-1213.

**List of changes**

Page 1, line 1-3:'Satellite observed indications of aerosol effects on warm cloud properties over Yangtze River Delta of China' was changed to 'Analysis of aerosol effects on warm clouds over the Yangtze River Delta from multi-sensor satellite observations'

Page 1, line 20-24: Text was changed to 'This study shows that the COT-CDR and CWP-CDR relationships are not unique, but affected by atmospheric aerosol loading. The relation between cloud properties and AOD also depends on the aerosol abundance, with a different behaviour for low and high AOD (i.e. AOD<0.35 and AOD>0.35). This applies to cloud droplet effective radius (CDR) and cloud fraction (CF), but not to cloud optical thickness (COT) and cloud top pressure (CTP).'

Page 1, line 27-30: Text was changed to 'Furthermore, separation of cases with either polluted dust or smoke aerosol shows that aerosol-cloud interaction (ACI) is stronger for clouds mixed with smoke aerosol than for clouds mixed with dust, which is ascribed to the higher absorption efficiency of smoke than dust.'

Page 1, line 31-39: Text was changed to 'high relative humidity favours larger cloud droplet particles and increases cloud formation, irrespective of vertical or horizontal level. Stable atmospheric conditions enhance cloud cover horizontally. However, unstable atmospheric conditions favour thicker and higher clouds. Dynamically, upward motion of air parcels can also facilitate the formation of thicker and higher clouds. Overall, the present study provides an understanding of the impact of aerosols on cloud properties over the YRD. In addition to the amount of aerosol particles (or AOD), evidence is provided that aerosol types and ambient environmental conditions need to be considered to understand the observed relationships between cloud properties and AOD.'

Page 3, line 23-25: Text was changed to 'Costantino and Breon (2013) and Jones et al. (2009), using MODIS data, found that the aerosol indirect effect is stronger for well-mixed clouds than for well-separated clouds (in well-mixed aerosol and cloud layers are physically interacting, as further explained in Section 2).'

Page 4, line 12-13: Text was added as: 'New insights into the changing cloud properties over a wide range of aerosol loadings, in particular in high AOD conditions,'

Page 4, line 25, 'Description of region interest' was changed to 'Description of the study region'

Page 6, line 11-15: Text was changed to 'Along with the Aqua satellites, CloudSat and CALIPSO (Cloud-Aerosol Lidar and Infrared Pathfinder Satellite Observations) are

flying in the so-called "A-train" constellation together with other NASA satellites (Stephens et al., 2002). CloudSat carries the CPR (Cloud Profiling Radar), i.e. the first satellite-based millimeter-wavelength cloud radar to detect the vertical information on different sized cloud droplets (Im et al., 2005).'

Page 6, line 24-26: Text was added as: 'The vertical resolution of the CALIOP layer product varies with altitude: 30 m for h =0 - 8.2 km, 60 m for h = 8.2 -20.2 km, and 180 m for h = 20.2 – 30.1 km, whereas the horizontal resolution is 5 km (Liu et al., 2009).'

Page 9, line 7: Text was added as:'3.1.1 Spatial and time series analysis of aerosol and cloud parameters'

Page 9, line 9-10: Text was added as:' Figure 2. Spatial distributions of AOD (a), CDR (b), CF (c), COT (d), CWP (e) and CTP (f) averaged over all years between 2007 and 2010.'

Page 9, line 12-17: Text was added as: 'The spatial variations of the aerosol and cloud properties over the study area, averaged over the years 2007-2010, are shown in Figure 2. We can see a decreasing north-south pattern in AOD in Figure 2a, with the highest values found in the northeast area. CDR behaves similar to AOD, except that the highest values are found in the northernmost area. Contrary to AOD, both COT and CWP show an increasing north-south pattern. Furthermore, the spatial distributions of COT and CWP are remarkably similar to each other.'

Page 10, line 2-3: Text was added as: 'Figure 3. Time series of the monthly averaged values of AOD (a), CDR (b), CF (c), COT (d), CWP (e) and CTP (f) for all months between 2007 and 2010. Month 1 is January.'

Page 10, line 4-14: Text was added as: 'Figure 3 shows time series of the monthly-averaged values for the AOD, CDR, COT, CWP, CF and CTP, calculated for each month during the four years 2007 - 2010. Both the monthly-averaged AOD and CDR are highest in June. December presents the lowest monthly-average for the AOD. Overall, the variations of the monthly-averaged COT and CWP are similar, with the lower values in the summer and the higher value in the winter. The monthly-averaged CF approaches its maximum values in Jan and June, while CTP shows two peaks in Feb and Sep. Note that CTP is plotted along the vertical axis from high to low. The monthly averages are determined from the numbers of samples presented in Table 2 for each parameter and each month between 2007 and 2010. Further, the availabilities of data for AOD and cloud properties are not the same for the whole acquisition period between 2007 and 2010. It indicates that not every CALIPSO shot has all the corresponding value for AOD, CDR, COT, CWP, CF or CTP, which will decrease the data sample size to some extent.'

Page 11, line 10-11: Text was added as: 'Prior to investigating the aerosol impact on warm cloud properties, a general analysis of cloud properties and the effect of aerosol loading on the relations between them are discussed below.'

Page 12, line 3-7: Text was changed to 'Student's t-test is used to determine whether two sets of data are significantly different from each other. The p-value is defined as the probability of obtaining a result equal to or "more extreme" than what was actually observed, when the null hypothesisis true. Marker * at the top right corner of R-value denotes statistically significant if p<0.05.'

Page 12, line 8-13: Text was added as: 'We first explored the response of CDR to the increasing AOD in mixed aerosol-cloud layers and found that CDR decreases with increasing AOD in moderately polluted conditions (AOD < 0.35). In polluted and heavily polluted conditions (AOD > 0.35), however, CDR increases with increasing AOD. Here we discriminate between moderately (AOD < 0.35), polluted (AOD >= 0.35 and AOD <=0.8) and heavily polluted (AOD >0.8) conditions. The threshold of 0.35 for AOD is chosen based on analysis presented below in section 3.2 where we compare the relation of cloud parameters and AOD in more detail.'

Page 12, line 24-27: Text was added as: 'Costantino and Bréon (2013) compared the CDR-COT relationship of mixed and separated aerosol-cloud layers and found an increase in the CDR with increasing COT, followed by a decrease with higher COT in both cases (mixed and separated aerosol-cloud layers). Compared to their study, we consider the effect of aerosol loading on the relationship between CDR and COT in both cases.'

Page 12, line 29: Text was changed to 'However, when different degrees of pollution are considered, Figure 4(d), we see a clear correlation between both parameters (R=0.78) in moderately polluted conditions, where CWP clearly increases with increasing CDR. In polluted and heavily polluted conditions the variation of CWP with increasing CDR is much weaker, R=0.31 for polluted conditions, and in heavily polluted conditions CWP decreases with increasing CDR (R=-0.33).'

Page 13, line 16-17: Text was changed to 'An explanation for this phenomenon is provided by Gao et al. (2014), i.e. clouds grow in the vertical and more drizzle is produced, so that the cloud liquid water path becomes larger. ns, and in heavily polluted conditions CWP decreases with increasing CDR (R=-0.33).'

Page 13, line 20-21: Text was added as: 'This observation is in good agreement with those of Costantino and Bréon (2013) that cloud water amount increases with increasing cloud optical thickness.'

Page 14, line 9-10: Text was added as: 'In this section we examine the responses of various cloud properties to the increasing AOD for well-separated and well-mixed

clouds, respectively.'

Page 15, line 16-18: Text was changed to 'Figure 6(b) shows that when aerosol and cloud layers physically interact, the CF shows a decrease with the increase of AOD in moderately polluted conditions, albeit with a low significance as indicated by the small correlation coefficient R, followed by an inverse pattern in polluted and heavily polluted conditions.'

Page 15, line 19-25: Text was added as: 'It could be explained as follows: Here, when aerosol and cloud layers are well-mixed, the absorption of solar radiation heats the mixed layer and reduces the cloud cover due to the quite high concentrations of the smoke particles over the YRD. This feedback would be balanced once the heating of the surface raises the surface temperature. It destabilizes the atmosphere resulting in vertical transport and thus enabling transfer of humidity from the surface to higher levels in the atmosphere. This effect increases cloudiness (Koren et al., 2008).'

Page 16, line 22-27: Text was added as: 'Based on the above findings, we conclude that for well-mixed clouds in the YRD, the CDR shows a decrease with an increasing AOD under moderately-polluted conditions, followed by an increase under polluted and heavily-polluted conditions due to the intense water vapour competition. The cloud cover behaves qualitatively similar to CDR in response to changing values of AOD. Meanwhile, cloud optical depth becomes smaller and cloud top pressure becomes larger with increasing AOD over the whole range of AOD values.'

Page 17, line 12-13: Text was changed to 'As with the CDR, the CF shows similar variation with the elevated AOD over the whole AOD range.'

Page 17, line 16 and Page 18, line 1-2: Text was added as: 'The slightly difference of fits comes from the different types of clouds are considered in different conditions. In fig 6, the clouds are not limited to single layer warm clouds, but also double layer warm clouds.'

Page 19, line 2: Text was changed to 'Smoke (fine absorbing particles) and polluted dust (coarse particles) aerosol are identified using the CALIOP classification. In addition, they have different efficiency for the absorption of sunlight.'

Page 19, line 7-8: Text was changed to 'Figure 8(a) shows that the CDR is, in general, larger in the presence of smoke aerosol than in the presence of dust.'

Page 19, line 14-16: Text was changed to 'The slope of linear regression of cloud optical thickness against AOD is much stronger in the presence of smoke aerosol than in the presence of dust, indicating that the ACI is stronger for smoke than for polluted dust.'

Page 20, line 7-12: Text was added as: 'Feingold et al. (2001) reported that the aerosol indirect effect depends highly on the aerosol hygroscopicity and pressure vertical velocity. Wang et al. (2014) demonstrated that the observed interaction between aerosol and cloud can be affected by the dynamical and thermodynamical processes in cloud systems. Therefore, to explore the meteorological impact on the interaction between aerosol and cloud observed over the YRD, we classify the data for various meteorological parameters, including RH (this section), LTS and PVV (Section 3.3.4).'

Page 21, line 11-17: Text was changed to 'Figure 9(c) shows that the cloud optical thickness decreases with increasing AOD in both conditions albeit with a low significance as indicated by the small correlation coefficient R. However, the cloud optical thickness is larger in high relative humidity conditions than in low relative humidity conditions for the entire AOD dataset. In contrast, the cloud top pressure is smaller in high relative humidity conditions than in low relative humidity conditions over the whole range of AOD values (Figure 9(d)). This implies that high relative humidity can promote the formation of thicker and higher clouds.'

Page 22, line 15-17: Text was changed to 'Figure 10(b) shows that the slope of linear regression of cloud fraction against AOD is much stronger for stable atmospheric conditions than for unstable atmospheric conditions in the heavily polluted conditions.'

Page 23, line 4-9: Text was changed to 'Figure 10(c) shows that the cloud optical thickness is larger in unstable atmospheric conditions than in stable atmospheric conditions. In contrast, the cloud top pressure is smaller in unstable atmospheric conditions than in stable atmospheric conditions for the whole range of AOD values (Figure 9(d)). This indicates that unstable atmospheric conditions can promote the formation of thicker and higher clouds and stable atmospheric conditions can enhance the cloud cover.'

Page 24, line 5-6: Text was changed to 'However, the impact of vertical velocity is weak in polluted and heavily polluted conditions.'

Page 24, line 12-16: Text was changed to 'Figure 11(c) shows that the cloud optical thickness is larger in the presence of upward motion of air parcels than for downward motion throughout the range of AOD. In contrast, the cloud top pressure is smaller in the presence of upward motion of air parcels than for downward motion (Figure 9(d)). This implies that upward motion of air parcels can be helpful for the formation of thicker and higher clouds.'

Page 24, line 17: Text was added as: '3.4 Error sources and uncertainties'

Page 24, line 18-28 and page 25, line 1-18: Text was added as: 'Caution is warranted

in investigating the satellite-derived relations between aerosol and cloud properties. Uncertainties in satellite data may results from assumptions on the aerosol size distribution used in the retrieval process, imperfect cloud detection resulting in residual clouds leading to high AOD values, effects of relative humidity on aerosol parameters, and dynamic effects (Yuan et al., 2008). Below we discuss several potential factors that may have affected the interaction between aerosols and clouds in our analysis.

Firstly, the correlation between AOD and cloud parameters may be influenced by aerosol size distributions (Small et al., 2011). Since the MODIS retrieval does not provide aerosol size information, it is better to explore the seasonal differences in the observed ACI due to the difference in aerosol emissions between the different seasons. However, the relatively low number of MODIS-CALIPSO coincidences limits the further binning of the data required to investigate this issue. Secondly, what it comes to the occurrence of cloud contamination in the AOD dataset, this is a universal and one of the most difficult problems in aerosol retrieval. Cloud detection is usually not perfect, so that undetected, or residual, clouds contaminate the retrieval area which leads to AOD overestimation and in turn affects the relation between aerosol and cloud properties (e.g. Sogacheva et al., 2017). A study by Mei et al. (2016), comparing their MERIS cloud mask with two independent data sets, shows that on the order of 70-90% of the cases are correctly classified as cloud free. This result is in good agreement with that from a dedicated study on a consistency between aerosol and cloud retrievals from the same instrument which showed that about 20% of the pixels may be mis-classified (Klueser, 2014). In this study, the samples with AOD values greater than 1.5 were excluded as a rough attempt to exclude cloud-contaminated AOD to reduce the uncertainty in the observed ACI. Thirdly, Feingold et al. (2003) reported that water vapour swelling increases the AOD. Sheridan et al. (2001) showed an important role of hygroscopic growth in determining the AOD for sea salt aerosols. The effect of humidity on the ACI has been discussed in Section 3.3.3. Finally, Young (1993) reported that ACI is influenced by dynamics through modifying radiative and thermodynamic heating. Jones et al. (2009) emphasized the importance of vertical mixing velocity in cloud formation and ACI as discussed in Sections 3.3.4 and 3.3.5. As reported by Yuan et al. (2010), the potential artefacts above mentioned do not seem to be the primary cause for the observed relationship between aerosol and cloud parameters. Further investigations are needed to fully analyse and explain the observed phenomena.'

Page 25, line 20-21: Text was changed to 'The high level of anthropogenic emissions in Eastern China render this area an important hotspot for studying how cloud microphysical properties are affected by anthropogenic aerosols (Ding et al., 2013).'

Page 26, line 3-4: Text was changed to 'The analysis of the COT-CDR and CWP-CDR relationships for well-mixed clouds shows that they are affected by the amount of aerosol.'

Page 26, line 12-15: Text was changed to 'Also, the analysis of the variation of CF with increasing AOD shows that CF varies with AOD in a way similar to that of CDR. This finding differs from those by Koren et al. (2008) and Small et al. (2011) who observed that an increase in the cloud cover with an increasing AOD, followed by a decrease with higher AOD.'

Page 26, line 19-30 and Page 27 line 1-2: Text was added as: 'In other words, we found that for well-mixed clouds over the YRD, the CDR becomes smaller with the increase of AOD in moderately polluted conditions which in principles in line with the Twomey effect, yet, the cloud fraction indicates a weak decrease which could be attributed only to the weak influence of evaporation caused by absorption of aerosols. On the other hand, in polluted and heavily polluted conditions, a reduced cloud coverage can result in more solar radiation reaching the surface, causing surface heating and thus raises the surface temperature, which then destabilizes the atmosphere. The resulting advection transports water vapour from the surface to higher levels in the atmosphere, therefore producing more cloud. Meanwhile, CDR becomes larger as a result of the stronger water vapour competition in polluted and heavily polluted conditions. The COT decreases with the increasing values of AOD throughout the AOD range due to the radiative effect and possible retrieval artefacts. The behaviour of CTP is consistent with that of COT, with the cloud getting thinner but with larger cover, so that CTP becomes larger with an increasing AOD.'

Page 27, line 8-13: Text was changed to 'In general, high relative humidity can promote the formation of larger cloud droplet particles and expand cloud formation, irrespective of vertical or horizontal level. With regard to LTS, stable atmospheric conditions can enhance the cloud cover horizontally. However, unstable atmospheric conditions can be helpful for the formation of thicker and higher clouds. Dynamically, upward motion of air parcels can also facilitate the formation of thicker and higher clouds.'

Page 27, line 16-17: Text was added as: 'In summary, this study will greatly help us to understand the mechanisms of aerosol-cloud interaction and ultimately of aerosol indirect effects over the YRD.'

**Analysis of aerosol effects on warm clouds over the Yangtze River Delta from multi-sensor satellite observations**

Yuqin Liu[1, 2, 3], Gerrit de Leeuw[1,3, 4], Veli-Matti Kerminen[3], Jiahua Zhang[1], Putian Zhou[3], Wei Nie[5], Ximeng Qi[5], Juan Hong[3],Yonghong Wang[3], Aijun Ding[5], Huadong Guo[1], Olaf Krüger[3], Markku Kulmala[3], Tuukka Petäjä[3]

1Institute of Remote Sensing and Digital Earth, Chinese Academy of Sciences, Beijing, China

2University of Chinese Academy of Sciences, Beijing, China

3Department of Physics, P.O. Box 64, 00014 University of Helsinki, Helsinki, Finland

4Finish Meteorological Institute, Climate Change Unit, P.O. Box 503, 00101 Helsinki, Finland

5Institute for Climate and Global Change Research & School of Atmospheric Sciences, Nanjing University, 210023 Nanjing, China

*Correspondence to*:J.H. ZHANG (jhzhang@ceode.ac.cn)

**Abstract.** Aerosol effects on low warm clouds over the Yangtze River Delta (YRD, East China) are examined using co-located MODIS, CALIOP and CloudSat observations. By taking the vertical locations of aerosol and cloud layers into account, we use simultaneously observed aerosol and cloud data to investigate relationships between cloud properties and the amount of aerosol particles (using aerosol optical depth, AOD, as a proxy). Also, we investigate the impact of aerosol types on the variation of cloud properties with AOD. Finally, we explore how meteorological conditions affect these relationships using ERA Interim Reanalysis data. This study shows that the COT-CDR and CWP-CDR relationships are not unique, but affected by atmospheric aerosol loading. The relation between cloud properties and AOD also depends on the aerosol abundance, with a different behaviour for low and high AOD (i.e. AOD<0.35 and AOD>0.35). This applies to cloud droplet effective radius (CDR) and cloud fraction (CF), but not to cloud optical thickness (COT) and cloud top pressure (CTP). COT is found to decrease when AOD increases, which may be due to radiative effects and retrieval artefacts caused by absorbing aerosol. Conversely, CTP tends to increase with elevated AOD, indicating that the aerosol is not always prone to expand the vertical extension. Furthermore, separation of cases with either polluted dust or smoke aerosol shows that aerosol-cloud interaction (ACI) is stronger for clouds mixed with smoke aerosol than for clouds mixed with dust, which is ascribed to the higher absorption efficiency of smoke than dust. The variation of cloud properties with AOD is analysed for various relative humidity (RH) and boundary layer thermodynamic and dynamic conditions, showing that high relative humidity favours larger cloud droplet particles and increases cloud formation, irrespective of vertical or horizontal level. Stable atmospheric conditions enhance cloud cover horizontally. However, unstable atmospheric conditions favour thicker and higher clouds. Dynamically, upward motion of air parcels can also facilitate the formation of thicker and higher clouds. Overall, the present study provides an understanding of the impact of aerosols on cloud properties over the YRD. In addition to the amount of aerosol particles (or AOD), evidence is provided that aerosol types and ambient environmental conditions need to be considered to understand 
[revised manuscript text omitted]
 insights into the changing cloud properties over a wide range of aerosol loadings, in particular in high AOD conditions, result from our focus on a systematic understanding of ACI from three perspectives: (1) well-mixed and well-separated clouds, (2) aerosol effects on properties of well-mixed clouds, (3) well-mixed clouds under different meteorological conditions.

The paper is organized as follows: section 2 describes the datasets used, data processing and the main analysis conducted to explore aerosol cloud interaction. Section 3 starts with a general description of aerosol and cloud properties and the effect of aerosol loading on the relations between them, followed by a description of aerosol effects on cloud properties (CDR, CF, COT and CTP). In the latter we discriminate between well-separated and well-mixed clouds. The focus will be on well-mixed clouds where ACI takes place, and aerosol types and meteorological factors are considered to better understand the possible mechanisms. Overall conclusions and discussions are presented in section 4.

**2. Methods**

**2.1 Description of the study region**

In this study, the Yangtze River Delta (YRD), covering the area 27°N-34°N and 115°E-122°E (Figure 1), was chosen in order to investigate the aerosol-induced variability in micro- and macrophysical properties of low warm clouds during four consecutive years (2007-2010). The YRD region was chosen because it

is representative for the continental East Asian subtropical climate. The marine monsoon subtropical climate for YRD is characterized by hot and humid summers and cool dry winters (Sundström et al., 2008; Zhang et al., 2010). The mean temperature in summer from 2007 to 2010 is about 27-28℃. Mean annual precipitation ranges from 1000 to 1400 mm and most precipitation occurs in spring and summer (Zhang et al., 2010; Cao et al., 2016).

The population density in the YRD is very high with intensive human activities in the region contributing to a very variable and complex aerosol composition. The YRD has been reported to be a major source region of both black carbon and sulfate (Wang et al., 2014; Andersson et al., 2015). In addition, other aerosol sources such as dust emissions render the interactions between aerosols and clouds complicated (Nie et al., 2014). The continental area of interest is characterized by a high level of anthropogenic emissions and is well suited for research related to the indirect effects of aerosols on cloud micro- and macro-physical properties.

[Figure]

Figure 1. Map of annual averaged MODIS/AQUA level 2 AOD for all years during the period from 2007 to 2010. The black rectangle (27°N-34°N and 115°E-122°E) indicates the Yangtze River Delta (YRD).

**2.2 Data sources**

The MODIS sensor, onboard the Aqua satellite, has a swath width of ~2300 km and multi-band spectral coverage (King et al., 2003). The MODIS/Aqua overpass time for the study area is around 13:30 local time, when continental warm clouds are likely to be well developed. Therefore MODIS/Aqua was selected as a data source to explore the ACI over this area. In this work, we used the MODIS Collection

5.1 AOD product (MOD04) derived from cloud-free pixels (resolution 500 m at nadir) and aggregated to a resolution of 10 km×10 km (Remer et al., 2005; Levy et al., 2010). The AOD over land is retrieved using three MODIS channels: 0.47, 0.66 and 2.13 μm (Remer et al., 2005). Cloud properties are retrieved using six spectral channels (King et al., 1998) at visible and near infrared wavelengths (i.e., 0.66, 0.86, 1.24, 1.64, 2.12 and 3.75 μm). Here, we used the AOD as a proxy for aerosol burden in our aerosol-cloud interaction analysis. The cloud properties used in this study, CDR, CWP, COT, cloud top pressure (CTP) and cloud phase infrared (CPI), were obtained from the Level 2 cloud product (MYD06) (King et al., 2003). Both these products MOD04 and MYD06 are in good agreement with ground-based remote sensingdata (Levy et al., 2010; Platnick et al., 2003). More detailed information on algorithms for the retrieval of aerosol and cloud properties is provided at http://modis-atmos.gsfc.nasa.gov.

Along with the Aqua satellites, CloudSat and CALIPSO (Cloud-Aerosol Lidar and Infrared Pathfinder Satellite Observations) are flying in the so-called "A-train" constellation together with other NASA satellites (Stephens et al., 2002). CloudSat carries the CPR (Cloud Profiling Radar), i.e. the first satellite-based millimeter-wavelength cloud radar to detect the vertical information on different sized cloud droplets (Im et al., 2005). The CPR is able to penetrate optically thick clouds and detect weak precipitating particles (Wang et al., 2013). In the present study we utilized the datasets CloudLayerBase and CloudLayerTop from 2B-CLDCLASS-LIDAR, the latest version (R04) of the CloudSat standard data products. The data are provided in the CPR spatial grid with vertical and horizontal resolutions of approximately 480 m and 1.4×1.8 km, respectively. CALIOP (Cloud-Aerosol Lidar with Orthogonal Polarization) on board CALIPSO is the first space-borne near-nadir polarization lidaroptimised for aerosol and cloud measurements (Winker et al., 2003). It is sensitive to optically thin clouds which could be missed by CPR (Wang et al., 2013). The datasets Layer_Base_Altitude and Layer_Top_Altitude retrieved from the CALIOP level-2 aerosol layers product (05kmALay) were used in the present study. Its footprint is very narrow, with a laser pulse diameter of 70 m on the ground. The vertical resolution of the CALIOP layer product varies with altitude: 30 m for h =0 - 8.2 km, 60 m for h = 8.2 -20.2 km, and 180 m for h = 20.2 – 30.1 km, whereas the horizontal resolution is 5 km (Liu et al., 2009). Combining CloudSat and CALIPSO observations has provided new insights into the vertical structure and microphysical properties of clouds (Matrosov, 2007).

The daily temperature at the 1000 hPa and 700 hPa levels, relative humidity at the 950hPa level and pressure vertical velocity at the 750 hPa level were obtained from ERA Interim Reanalysis data. The

daily ERA Interim Reanalysis contains global meteorological conditions with 0.125°×0.125° grids and a 37 level vertical resolution (1000-1 hPa) every six hours (00:00, 06:00, 12:00, 18:00 UTC) (http://apps.ecmwf.int/datasets/data/interim-full-daily/). The reanalysis data were used for the closest collocation with the satellite overpass time over the study area.

5    Table 1. Level 2 MODIS, CALIOP, CALIOP/CPR and ERA Interim products used to characterize aerosol and cloud properties.

| Product | Dataset | Horizontal resolution | Data source |
|---------|---------|-----------------------|-------------|
| Aerosol(MYD04 Level 2 Collection 5) | Optical_Depth_Land_And_Ocean | 10 km | MODIS |
| Cloud(MYD06 Level 2 Collection 5) | Cloud_Effective_Radius | 1 km | |
| | Cloud_Water_Path | 1 km | |
| | Cloud_Phase_Infrared_Day | 5 km | |
| | Cloud_TOP_Pressure_Day | 5 km | |
| | Cloud_Fraction_Day | 5 km | |
| | Cloud_Optical_Thickness | 1 km | |
| Cloud(2B-CLDCLASS-LIDAR) | CloudLayerBase | 2.5 km | CALIOP/CPR |
| | CloudLayerTop | 2.5 km | |
| Aerosol(05kmALay) | Layer_Top_Altitude | 5 km | CALIOP |
| | Layer_Base_Altitude | 5 km | |
| | Cloud_Aerosol_Discrimination | 5 km | |
| | Feature_Classification_Flags | 5 km | |
| ERA Interim | Temperature (700hPa, 1000hPa) | 0.125° | ECMWF |
| | Relative humidity (950hPa) | 0.125° | |
| | Pressure vertical velocity (750hPa) | 0.125° | |

**2.3 Data processing**

The MODIS/AQUA, CALIOP/CALIPSO and CPR/CLOUDSAT satellites are part of the A-Train constellation and observe the same scene on Earth within one to two minutes (Stephens et al., 2002).

10    Therefore, time-coincidence of retrievals is assured when the datasets are extracted for the same date. Meteorological properties retrieved from the 06:00 UTC ERA Interim datasets were used here as the "A-train" satellites constellation overpasses the region of interest at about 13:30 local time (05:30 UTC). We aggregated CDR, COT and CWP (1 km × 1 km) to a resolution of 5 km × 5km to match the along-track resolution of CALIOP (5 km × 5 km), while CTP, CF and CPI were directly applied for the

15    analysis since all of them are at a 5 km × 5km spatial resolution.

Aerosol properties are only retrieved for strictly cloud-free pixels as determined by the application of a cloud-detection scheme. However, cloud detection schemes are not perfect and some residual clouds may remain undetected resulting in high AOD (Kaufman et al., 2005b). Another potential source of error could be the misclassification of high AOD areas, such as in the presence of desert dust or very high pollution levels, as clouds. To reduce a possible over-estimation of AOD, cases with AOD greater than 1.5 were excluded from further analysis. In this paper, we focused on warm clouds with CTP larger than 700 hPa and CWP lower than 200 g m$^{-2}$, as most aerosols exist in the lower troposphere (Michibata et al. 2014). In addition, only cases with CPI = 1 (liquid water cloud) were included. When CALIOP detected the presence of aerosol, we averaged the MODIS aerosol retrievals within a radius of 50 km from the CALIOP target. Likewise, we averaged the MODIS cloud retrievals within a radius of 5 km from the CALIOP target. For meteorological properties, we chose the value of the footprint that is nearest to the CALIOP target. MODIS, CALIOP, and CPR datasets are listed in Table 1.

A quantitative relationship between aerosol optical depth and cloud properties has been documented in previous studies (Sporre et al., 2014; Meskhidze and Nenes, 2010; Koren et al., 2005, Saponaro et al., 2017). However, the relative vertical positions of aerosol and cloud layers contribute to the uncertainty in this relationship. Following the method by Costantino and Breon (2013), we considered the aerosol and cloud layers to be physically interacting (well mixed) when the vertical distance between bottom (top) of the aerosol layer and the top (bottom) of a cloud layer was smaller than 100 m. Coincident samples with a vertical distance larger than 750 m were assumed to be "well separated". Coincident samples with a distance between 100 and 750 m were defined as "uncertain". The uncertain cases, as identified using the information from CloudSat, were excluded from further analysis in this study. Cloud types were identified as single-, double- and multi-layer clouds using the cloud layer information at each point. Single-, double- and multi-layer cloud samples accounted for 59 %, 30% and 11 % of the total samples, respectively. Using the highest occurrence frequency (OF) of aerosol type below 10 km altitude at each point, the aerosol type of highest OF was defined following the Feature_Classification_Flags derived from CALIOP.

Meteorological and aerosol impacts on cloud macrophysics and microphysics are found to be tightly intermingled (Stevens and Feingold, 2009). In an attempt to isolate aerosol effects, the meteorological effects on clouds were explored in a statistical sense. Meteorological properties used here include relative humidity (RH), lower tropospheric stability (LTS) and pressure vertical velocity (PVV). LTS is defined

as the difference in potential temperature between the free troposphere (700hpa) and the surface, which is representative of typical thermodynamic conditions (Klein and Hartmanm, 1993). RH, LTS and PVV have been suggested to affect aerosol and cloud interaction (Gryspeerdt and Partridge, 2014; Small et al., 2011). A positive LTS is associated with a stable atmosphere in which vertical mixing is prohibited; negative PVV indicates a local upward motion of air parcels.

**3. Results and Discussions**

**3.1 Overall aerosol and cloud characteristics**

**3.1.1 Spatial and time series analysis of aerosol and cloud parameters**

[Figure]

Figure 2. Spatial distributions of AOD (a), CDR (b), CF (c), COT (d), CWP (e) and CTP (f) averaged over all years between 2007 and 2010.

The spatial variations of the aerosol and cloud properties over the study area, averaged over the years 2007-2010, are shown in Figure 2. We can see a decreasing north-south pattern in AOD in Figure 2a, with the highest values found in the northeast area. CDR behaves similar to AOD, except that the highest values are found in the northernmost area. Contrary to AOD, both COT and CWP show an increasing north-south pattern. Furthermore, the spatial distributions of COT and CWP are remarkably similar to each other.

[Figure]

Figure 3. Time series of the monthly averaged values of AOD (a), CDR (b), CF (c), COT (d), CWP (e) and CTP (f) for all months between 2007 and 2010. Month 1 is January.

Figure 3 shows time series of the monthly-averaged values for the AOD, CDR, COT, CWP, CF and CTP,

5   calculated for each month during the four years 2007 - 2010. Both the monthly-averaged AOD and CDR are highest in June. December presents the lowest monthly-average for the AOD. Overall, the variations of the monthly-averaged COT and CWP are similar, with the lower values in the summer and the higher value in the winter. The monthly-averaged CF approaches its maximum values in Jan and June, while CTP shows two peaks in Feb and Sep. Note that CTP is plotted along the vertical axis from high to low.

10  The monthly averages are determined from the numbers of samples presented in Table 2 for each parameter and each month between 2007 and 2010. Further, the availabilities of data for AOD and cloud properties are not the same for the whole acquisition period between 2007 and 2010. It indicates that not every CALIPSO shot has all the corresponding value for AOD, CDR, COT, CWP, CF or CTP, which will decrease the data sample size to some extent.

Table 2. The sample sizes of all months for each parameter

| Parameters | Jan | Feb | Mar | Apr | May | Jun | Jul | Aug | Sep | Oct | Nov | Dec | Total |
|---|---|---|---|---|---|---|---|---|---|---|---|---|---|
| AOD | 5428 | 3332 | 3892 | 4704 | 5598 | 3638 | 5944 | 6630 | 4306 | 6728 | 6110 | 6400 | 62710 |
| CDR | 794 | 669 | 365 | 679 | 714 | 872 | 1228 | 2013 | 1514 | 1281 | 895 | 582 | 11606 |
| COT | 886 | 747 | 392 | 732 | 748 | 915 | 1298 | 2072 | 1539 | 1329 | 967 | 627 | 12232 |
| CWP | 1226 | 1125 | 620 | 1310 | 1226 | 1245 | 1490 | 2187 | 1929 | 1715 | 1261 | 867 | 16201 |
| CF | 1398 | 994 | 537 | 955 | 993 | 1065 | 1671 | 2650 | 1996 | 1811 | 1373 | 1119 | 16562 |
| CTP | 1398 | 994 | 537 | 955 | 993 | 1065 | 1671 | 2650 | 1996 | 1811 | 1373 | 1119 | 16562 |

**3.1.2 Variation of COT and CWP with CDR**

[Figure]

Figure 4. Scatterplots of cloud parameters versus CDR in well-mixed aerosol-cloud layers: (a) COT and (b) CWP, both for all data; (c) COT and (d) CWP, both for data grouped by moderately polluted (in blue), polluted (in green) and heavily polluted (in red) atmospheric conditions. Here moderately polluted refers to AOD <0.35, polluted refers to 0.35<=AOD <= 0.8 and heavily polluted refers to AOD>0.8. The lines present the least squares fits and the resulting relations are presented in each figure. The number of data samples is also reported in the figure (and following figures).

Prior to investigating the aerosol impact on warm cloud properties, a general analysis of cloud properties and the effect of aerosol loading on the relations between them are discussed below. The

overall statistical relations between the cloud parameters used in this study are derived from the scatterplots shown in Figure 4. All CDR, COT, CTP and CWP data shown in Figure 4 (and later figures) are averaged over AOD bins, from 0.05 to 1.5 with a step of 0.02 on a log-log scale. Student's t-test is used to determine whether two sets of data are significantly different from each other. The p-value is defined as the probability of obtaining a result equal to or "more extreme" than what was actually observed, when the null hypothesisis true. Marker * at the top right corner of R-value denotes statistically significant if $p<0.05$.

We first explored the response of CDR to the increasing AOD in mixed aerosol-cloud layers and found that CDR decreases with increasing AOD in moderately polluted conditions (AOD < 0.35). In polluted and heavily polluted conditions (AOD > 0.35), however, CDR increases with increasing AOD. Here we discriminate between moderately (AOD < 0.35), polluted (AOD >= 0.35 and AOD <=0.8) and heavily polluted (AOD >0.8) conditions. The threshold of 0.35 for AOD is chosen based on analysis presented below in section 3.2 where we compare the relation of cloud parameters and AOD in more detail. Figure 4(a) shows a scatterplot of COT vs CDR for well-mixed clouds. The correlation between these parameters is negative, i.e. COT decreases with CDR, with a correlation coefficient equal to -0.47. Figure 4(c) shows the same data but distinction is made between data points with in moderately polluted, polluted and heavily polluted conditions. For this dataset, COT increases with an increasing CDR at moderately polluted conditions. In contrast, for heavily polluted conditions COT shows a decrease with an increasing CDR. This may indicate the existence of intense competition between the aerosol particles for water vapour where the larger droplets are more prone for condensation of water vapour than smaller ones, and thus grow to larger sizes. This results in a shift of the droplet spectrum to larger sizes due to the increase of CDR accompanied by a decrease of COT (Wang et al., 2015). The data for the three different AOD cases show that the relationship between CDR and COT is not unique and depends on the aerosol abundance. Costantino and Bréon (2013) compared the CDR-COT relationship of mixed and separated aerosol-cloud layers and found an increase in the CDR with increasing COT, followed by a decrease with higher COT in both cases (mixed and separated aerosol-cloud layers). Compared to their study, we consider the effect of aerosol loading on the relationship between CDR and COT in both cases.

Figure 4(b) shows a weak correlation between CWP and CDR for well-mixed cloud layers, with a correlation coefficient equal to -0.15. However, when different degrees of pollution are considered, Figure 4(d), we see a clear correlation between both parameters (R=0.78) in moderately polluted

conditions, where CWP clearly increases with increasing CDR. In polluted and heavily polluted conditions the variation of CWP with increasing CDR is much weaker, R=0.31 for polluted conditions, and in heavily polluted conditions CWP decreases with increasing CDR (R=-0.33).

**3.1.3 Variation of COT and CWP with cloud top height**

[Figure]

Figure 5. Scatterplots of cloud parameters in well-mixed aerosol cloud layers for all data: (a) CTP versus COT, (b) CTP versus CWP, (c) CWP and COT. The lines present the least squares fits and the resulting relations are presented in each figure.

CTP is generally used as a measure of cloud top height (CTH), with higher CTP implying a lower CTH.

10 Figure 5(a) shows a positive correlation between CTP and COT, implying the occurrence of higher clouds with an increasing COT, which is consistent with the general understanding of aerosol-cloud interactions. Note that here and in the following figures, CTP is plotted along the vertical axis from high to low, i.e. decreasing CTP indicates increasing CTH, and positive correlations between CTP and other cloud parameters indicate that an increase in these parameters corresponds to a higher CTH. Figure 5(b)

15 shows a positive correlation between CTP and CWP, which again implies that clouds are higher as CWP increases. An explanation for this phenomenon is provided by Gao et al. (2014), i.e. clouds grow in the vertical and more drizzle is produced, so that the cloud liquid water path becomes larger. Figure 5(c) shows the relation between CWP and COT. The CWP increases with the increase of COT, which is in good agreement with the aerosol second indirect effect hypothesis that the precipitation suppression can

20 increase CWP and possibly further increase COT. This observation is in good agreement with those of Costantino and Bréon (2013) that cloud water amount increases with increasing cloud optical thickness.

**3.2 Difference between separated and mixed conditions**

[Figure]

Figure 6. Scatterplots of cloud parameters versus AOD over YRD on log-log scale for cases of separated (blue) and mixed (red) aerosol-cloud layers, (a) CDR versus AOD, (b) CF versus AOD, (c) COT versus AOD and (d) CTP versus AOD. The lines present the least squares fits and the resulting relations are presented in each figure. Error bars represent the confidence level of the mean cloud parameters' value for each AOD bin, i.e. the statistical uncertainties, expressed as $\sigma/(n-2)$ , where n is the number of cases within the AOD bin and $\sigma$ is the standard deviation of cloud properties.

In this section we examine the responses of various cloud properties to the increasing AOD for well-separated and well-mixed clouds, respectively. Figure 6 shows relations between cloud parameters (CDR, CF, COT, CTP) and AOD for both separated and mixed conditions. The strength of the interaction between cloud properties and AOD is quantified here as the slope of the line describing the relation between cloud parameters and AOD, on a log-log scale, as obtained by linear regression. In figure 6(a), CDR shows a negative relation with AOD in moderately polluted conditions when aerosol and cloud layers are mixed, which is in good agreement with Twomey's theory (Twomey, 1977). We note that, due to the limited number of data points in the dataset with AOD < 0.35, the present work does not allow selecting conditions with a constant CWP. Following, e.g., Costantino and Breon (2010; 2013) and

Wang (2015) we use all available data together. In polluted and heavily polluted conditions, however, CDR increases with increasing AOD, suggesting some sort of saturation in aerosol-cloud interactions when AOD approaches 0.35. This value for the tipping point (0.35) is close to the value of 0.4 reported by Feingold et al. (2001). As discussed earlier, Feingold et al. (2001) proposed three primary responses

5    of CDR to the aerosol loading. We consider the fact that CDR increases with an increase in AOD when AOD loading exceeds 0.35 as "anti-Twomey effect". The positive relation between CDR and AOD may be similar to that described by Feingold et al. (2001), case 3 (see above), i.e. due to intense vapour competition the smaller droplets evaporate as the number of particles continues to increase. It may also be that only a subset of aerosol particles is activated when not enough vapour is available, and once

10   activated they continue to grow faster, thus preventing water vapour from condensing onto smaller aerosol particles that are less susceptible to activation, resulting in the increase of CDR.

Figure 6a also shows that in well-separated cloud layers CDR varies much less with AOD irrespective of whether the AOD is relatively low or high. Such a weaker variation can be attributed to the fact that no aerosols are subjected to cloud microphysical process since there are no physical interactions between

15   aerosol and cloud layers.

Figure 6(b) shows that when aerosol and cloud layers physically interact, the CF shows a decrease with an increasing AOD in moderately polluted conditions, albeit with a low significance as indicated by the small correlation coefficient R, followed by an inverse patternin polluted and heavily polluted conditions. This outcome is not in agreement with the findings of Koren et al. (2008) and Small et al. (2011). It could

20   be explained as follows: Here, when aerosol and cloud layers are well-mixed, the absorption of solar radiation heats the mixed layer and reduces the cloud cover due to the quite high concentrations of the smoke particles over the YRD. This feedback would be balanced once the heating of the surface raises the surface temperature. It destabilizes the atmosphere resulting in vertical transport and thus enabling transfer of humidity from the surface to higher levels in the atmosphere. This effect increases

25   cloudiness (Koren et al., 2008). Conversely, CF shows an increasing pattern with an increasing AOD for the whole AOD dataset in well-separated cloud layers. This increase might be due to absorbing aerosols interacting with incoming solar radiation above the cloud layer (Costantino and Bréon, 2013). In this process, absorbing aerosols above cloud top may heat the aerosol layer and cool the surface, thereby stabilizing the boundary layer and maintaining a moist boundary layer. In addition, scattering aerosol

reduces the amount of solar light reaching the surface. This combination of two effects suppresses cloud vertical development and increase the low cloud cover.

The COT has a negative correlation with AOD in both conditions, as shown in Figure 6(c). There are two effects that may contribute to this negative relationship. On one hand, the evaporation of cloud droplets caused by locally absorbing aerosol makes clouds thinner, which is a radiative effect. On the other hand, the presence of absorbing aerosol may influence the satellite-retrieved COT because it can absorb radiation and thus reduce the cloud reflectance measured by the sensors on the satellite (Meyer et al., 2013; Li et al., 2014; Meyer et al., 2015; Hoeve et al., 2011). Meyer et al. (2013) reported that adjusting for above-cloud aerosol attenuation can increase the retrieved regional mean COT by roughly 18% for polluted marine boundary layer clouds. Li et al. (2014) also found that,due to absorbing aerosols in the heart of the Yangtze Delta region, satellite observations tend to underestimate COT. The radiative effect and retrieval uncertainty could be the important factors for the decrease of COT with increasing AOD, as suggested by Hoeve et al. (2011) and Alam et al. (2014). These authors reported similar results on the decrease of COT with increasing AOD, which may result from the measured reflectance from a cloud top at wavelengths in the visible being smaller than expected due to absorbing aerosols.

The relationship between CTP and AOD has been plotted in Figure 6(d). There is a positive correlation between CTP and AOD, which is contradicting the general understanding that high aerosol loading will result in an increase of cloud lifetime and higher cloud top. The positive relation between CTP and AOD has an implication that higher aerosol abundance is not always accompanied by smaller cloud top pressure. This suggests that the primary effect of aerosol is not always to produce taller and more convective clouds in some cases (Rennóet al., 2013).

Based on the above findings, we conclude that for well-mixed clouds in the YRD, the CDR shows a decrease with an increasing AOD under moderately-polluted conditions, followed by an increase under polluted and heavily-polluted conditions due to the intense water vapour competition. The cloud cover behaves qualitatively similar to CDR in response to changing values of AOD. Meanwhile, cloud optical depth becomes smaller and cloud top pressure becomes larger with increasing AOD over the whole range of AOD values.

**3.3 Case of mixed aerosol-cloud layers**

**3.3.1 ACI for single-layer mixed clouds**

[Figure]

Figure 7. Scatterplots of cloud parameters versus AOD over YRD on log-log scale for mixed aerosol-single layer clouds, (a) CDR, (b) CF, (c) COT and (d) CTP. The lines present the least squares fits and the resulting relations are presented in each figure. The error bars indicate the statistical uncertainties as in Fig. 6.

Well-mixed clouds show a stronger relation between aerosol and cloud properties than separated clouds, as shown above. From here on, we will focus on potential aerosol indirect effects on well-mixed warm clouds as defined above. Relations between CDR, CF, COT and CTP with AOD will be explored in this section. Figure 7 shows the variation of single layer cloud properties with AOD when aerosol and cloud layers are mixed. The relation between CDR and AOD changes from negative for AOD < 0.35 to positive for AOD > 0.35 (Figure 7a). As with the CDR, the CF shows similar variation with the elevated AOD over the whole AOD range. Figure 7(c) shows COT is negatively associated with increasing of AOD. In contrast, CTP decreases with increasing AOD (Figure 7d), i.e. cloud top height increases. In general, the characteristics for cases of mixed aerosol-single layer warm clouds (Figure 7) are quite similar to the case of mixed aerosol-warm clouds (Figure 6). The slightly difference of fits comes from

the different types of clouds are considered in different conditions. In fig 6, the clouds are not limited to single layer warm clouds, but also double layer warm clouds.

**3.3.2 Influence of aerosol type on ACI**

[Figure]

Figure 8. Scatterplots of cloud parameters versus AOD over YRD on log-log scale for cases of mixed dust aerosol-cloud layers (blue) and mixed smoke aerosols-cloud layers (red), (a) CDR, (b) CF, (c) COT and (d) CTP. The lines present the least squares fits and the resulting relations are presented in each figure. The error bars indicate the statistical uncertainties as in Fig. 6.

Eastern China is a region with high concentrations of sulfate, dust, black carbon and other carbonaceous aerosols. In heavily polluted areas, dust aerosols become coated with hygroscopic material, making them effective CCN (Levin et al., 1996; Satheesh et al., 2006). Especially, there are high emissions of smoke by strawburning in summertime. Aerosol - cloud interaction is strongly dependent on the aerosol types, their size distribution and the vertical variation of these, as well as ambient environmental conditions (Patra et al., 2005; Matsui et al., 2006; Dusek et al., 2008; Yuan et al., 2008). Thus, aerosol species are indicative of causal microphysical and radiative effects. Different aerosol types may reveal different patterns of ACI. Here, polluted dust (accounting for 34%) and smoke aerosol (accounting for 38%),

which are the two main aerosol types occurring in the YRD, are chosen to investigate the variation of cloud parameters with AOD. Smoke (fine absorbing particles) and polluted dust (coarse particles) aerosol are identified using the CALIOP classification. In addition, they have different efficiency for the absorption of sunlight.

5    Figure 8 shows the variation of cloud parameters with AOD over the YRD, where data points for mixed polluted dust-warm clouds and mixed smoke aerosols-warm clouds are indicated with different colours. Figure 8(a) shows that the CDR is, in general, larger in the presence of smoke aerosol than in the presence of dust. Meanwhile, the cloud fraction is smaller in the presence of smoke, as shown in Figure 8(b). This can be due to the greater efficiency of smoke aerosol particles for the absorption of sunlight

10    than that of dust, resulting in local warming in the presence of smoke aerosol which in turn leads to evaporation of water and thus an increase in small droplets or even complete evaporation of cloud droplets and thus a reduction of cloud cover. Figure 8(c) shows that the cloud optical thickness decreases with increasing AOD for both aerosol types albeit with a low significance as indicated by the small correlation coefficient R. The slope of linear regression of cloud optical thickness against AOD is much

15    stronger in the presence of smoke aerosol than in the presence of dust, indicating that the ACI is stronger for smoke than for polluted dust. In addition to those mentioned, one factor which probably also contributes to the observed difference between effects of smoke and polluted dust is that dust does not absorb sunlight at 0.86µm (Kaufman et al., 2005). Figure 8(d) shows that the slope of linear regression of cloud top pressure against AOD is much stronger for smoke aerosol than that for polluted aerosol, with a

20    correlation coefficient equal to 0.36. Both these results may be due to the higher absorption efficiency of smoke (Small et al., 2011).

**3.3.3 Influence of relative humidity on ACI**

[Figure]

Figure 9. Scatterplots of cloud parameters versus AOD over YRD on log-log scale for cases of low RH (31%) condition (blue) and mixed aerosol-cloud layers under high RH (91%) condition (red), (a) CDR, (b) CF, (c) COT and (d) CTP. The lines present the least squares fits and the resulting relations are presented in each figure. The error bars indicate the statistical uncertainties as in Fig. 6.

Feingold et al. (2001) reported that the aerosol indirect effect depends highly on the aerosol hygroscopicity and pressure vertical velocity. Wang et al. (2014) demonstrated that the observed interaction between aerosol and cloud can be affected by the dynamical and thermodynamical processes in cloud systems. Therefore, to explore the meteorological impact on the interaction between aerosol and cloud observed over the YRD, we classify the data for various meteorological parameters, including RH (this section), LTS and PVV (Section 3.3.4).

Relative humidity (RH) is one of the main factors affecting aerosol particle size and cloud formation. For instance, high RH at cloud base has been reported to affect the relation between aerosol particles and cloud properties (Small et al., 2011). Thus, effects of RH need to be accounted for in aerosol-cloud interaction studies, as reported in the literature (Jeong et al., 2007; Loeb and Manalo-Smith, 2005; Quaas et al., 2010).

The cloud properties versus AOD relationships are classified by RH (at 950hPa) in three equally sized subsets and the mean RH values for each subset are calculated. In figure 9 we show cloud properties as function of AOD for only the lowest RH (31%), representing dry conditions, and the highest RH (91%, above the deliquescence point of ambient particles). Figure 9(a) shows that the CDR is larger in high

5 relative humidity conditions than in low relative humidity conditions, irrespective of the AOD. It is likely that hygroscopic aerosols grow in size caused by condensation of water vapour (Hanel, 1976; Feingold et al., 2003). The increasing RH further increases the probability of the cloud droplet activation and growth of existing cloud droplets as well (Jones et al., 2009). This indicates that high relative humidity conditions can help the formation of larger cloud droplets due to a higher water vapour content in the

10 atmosphere. The cloud fraction is much larger in high relative humidity conditions than in low relative humidity conditions, as shown in Figure 9(b). Figure 9(c) shows that the cloud optical thickness decreases with increasing AOD in both conditions, albeit with a low significance as indicated by the small correlation coefficient R. However, the cloud optical thickness is larger in high relative humidity conditions than in low relative humidity conditions for the entire AOD dataset. In contrast, the cloud top

15 pressure is smaller in high relative humidity conditions than in low relative humidity conditions over the whole range of AOD values (Figure 9(d)). This implies that high relative humidity can promote the formation of thicker and higher clouds.

**3.3.4 Influence of boundary layer thermodynamics and dynamics on ACI**

[Figure]

Figure 10. Scatterplots of cloud parameters versus AOD over YRD on log-log scale for cases of low LTS condition (blue) and mixedaerosol-cloud layers under high LTS condition (red), (a) CDR, (b) CF, (c) COT and (d) CTP. The lines present the least squares fits and the resulting relations are presented in each figure. The error bars indicate the statistical uncertainties as in Fig. 6.

The LTS is an indicator for the mixing state of the atmospheric layer adjacent to the surface. It describes to some extent the atmosphere's tendency to promote or suppress vertical motion (Medeiros and Stevens, 2011), which in turn affects cloud properties (Klein and Hartmann, 1993).

Figure 10 shows cloud properties as function of AOD for two different LTS conditions: low LTS, with a mean value equal to 10.11 representing an unstable atmosphere; and high LTS, with a mean value equal to 20.47 representing a stable atmosphere. Figure 10(a) shows that the CDR is larger in unstable atmospheric conditions than in stable conditions, irrespective of the AOD. This indicates that in unstable atmospheric conditions the cloud droplets are larger, which may be due to stronger interaction between aerosols and clouds as a result of better vertical mixing of water vapour. Figure 10(b) shows that the slope of linear regression of cloud fraction against AOD is much stronger for stable atmospheric conditions than for unstable atmospheric conditions in the heavily polluted conditions. This

demonstrates that stable atmospheric conditions can promote the formation of a cloud (Small, et al., 2011). A high LTS indicates a strong inversion, which prevents vertical mixing and cloud vertical extent, maintaining a well-mixed and moist boundary layer and providing an environment which favours the development of a low cloud cover. ==Figure 10(c) shows that the cloud optical thickness is larger in==

5      ==unstable atmospheric conditions than in stable atmospheric conditions. In contrast, the cloud top pressure is smaller in unstable atmospheric conditions than in stable atmospheric conditions for the whole range of AOD values (Figure 9(d)). This indicates that unstable atmospheric conditions can promote the formation of thicker and higher clouds and stable atmospheric conditions can enhance the cloud cover.==

[Figure]

Figure 11. Scatterplots of cloud parameters versus AOD over YRD on log-log scale for cases of PVV<0 condition (blue) and mixed aerosol-cloud layers under high PVV>0 condition (red), ==(a) CDR, (b) CF, (c) COT and (d) CTP. The lines present the least squares fits and the resulting relations are presented in each figure.== The error bars indicate the statistical uncertainties as in Fig. 6.

15     The PVV, a measure of dynamic convection strength, is very important for cloud formation. In particular, the vertical velocity can be used to determine whether a certain region may be susceptible to cloud development or not. That is, the presence of upward motion, as indicated by negative PVV, can enhance

ACI as it makes the ambient environment favourable for cloud formation, and vice versa (Jones et al., 2009).

Figure 11(a) shows that in moderately polluted condition the CDR is larger in the presence of upward motion of air parcels than for downward motion. This observation indicates that the upward motion of air parcels can promote the formation of larger cloud droplets, thus enhancing ACI. However, the impact of vertical velocity is weak in polluted and heavily polluted conditions. Figure 11(b) shows that the cloud fraction is larger in the presence of upward motion of air parcels than for downward motion of air parcels when AOD is greater than 0.35. This indicates that the upward motion of air parcels can favour cloud development and increase cloud cover in heavily polluted conditions. The phenomenon is not obvious when AOD is smaller than 0.35. These results emphasize the importance of vertical velocity when estimating the potential aerosol effect on cloud droplet effective radius and cloud fraction in polluted conditions. Figure 11(c) shows that the cloud optical thickness is larger in the presence of upward motion of air parcels than for downward motion throughout the range of AOD. In contrast, the cloud top pressure is smaller in the presence of upward motion of air parcels than for downward motion (Figure 9(d)). This implies that upward motion of air parcels can be helpful for the formation of thicker and higher clouds.

**3.4 Error sources and uncertainties**

Caution is warranted in investigating the satellite-derived relations between aerosol and cloud properties. Uncertainties in satellite data may results from assumptions on the aerosol size distribution used in the retrieval process, imperfect cloud detection resulting in residual clouds leading to high AOD values, effects of relative humidity on aerosol parameters, and dynamic effects (Yuan et al., 2008). Below we discuss several potential factors that may have affected the interaction between aerosols and clouds in our analysis.

Firstly, the correlation between AOD and cloud parameters may be influenced by aerosol size distributions (Small et al., 2011). Since the MODIS retrieval does not provide aerosol size information, it is better to explore the seasonal differences in the observed ACI due to the difference in aerosol emissions between the different seasons. However, the relatively low number of MODIS-CALIPSO coincidences limits the further binning of the data required to investigate this issue. Secondly, what it

comes to the occurrence of cloud contamination in the AOD dataset, this is a universal and one of the most difficult problems in aerosol retrieval. Cloud detection is usually not perfect, so that undetected, or residual, clouds contaminate the retrieval area which leads to AOD overestimation and in turn affects the relation between aerosol and cloud properties (e.g. Sogacheva et al., 2017). A study by Mei et al. (2016), comparing their MERIS cloud mask with two independent data sets, shows that on the order of 70-90% of the cases are correctly classified as cloud free. This result is in good agreement with that from a dedicated study on a consistency between aerosol and cloud retrievals from the same instrument which showed that about 20% of the pixels may be mis-classified (Klueser, 2014). In this study, the samples with AOD values greater than 1.5 were excluded as a rough attempt to exclude cloud-contaminated AOD to reduce the uncertainty in the observed ACI. Thirdly, Feingold et al. (2003) reported that water vapour swelling increases the AOD. Sheridan et al. (2001) showed an important role of hygroscopic growth in determining the AOD for sea salt aerosols. The effect of humidity on the ACI has been discussed in Section 3.3.3. Finally, Young (1993) reported that ACI is influenced by dynamics through modifying radiative and thermodynamic heating. Jones et al. (2009) emphasized the importance of vertical mixing velocity in cloud formation and ACI as discussed in Sections 3.3.4 and 3.3.5. As reported by Yuan et al. (2010), the potential artefacts above mentioned do not seem to be the primary cause for the observed relationship between aerosol and cloud parameters. Further investigations are needed to fully analyse and explain the observed phenomena.

**4. Conclusions**

The high level of anthropogenic emissions in Eastern China render this area an important hotspot for studying how cloud microphysical properties are affected by anthropogenic aerosols (Ding et al., 2013). Based on the near-simultaneous aerosol and cloud retrievals provided by MODIS, CALIOP and CloudSat, together with the ERA Interim Reanalysis data, we investigated the effect of aerosols, with AOD used as a proxy for the aerosol loading, on micro-physical and macro-physical cloud properties over the Yangtze River Delta for the years 2007 to 2010. In terms of the relative heights of aerosol and cloud layers, well-mixed and separated clouds were defined. A statistical analysis was used to examine the aerosol effects on cloud properties for these two cases. Besides the aerosol impact on CDR, CF, COT and CTP, also the influence of environmental conditions, such as RH, LTS and PVV, on the relation

between cloud properties and AOD was studied. In addition, the impact of two different aerosol types, dust and smoke, was explored.

The analysis of the COT-CDR and CWP-CDR relationships for well-mixed clouds indicated that they are affected by the aerosol loading. A statistical analysis of the relation between CWP and COT showed an increase in CWP with an increasing COT, which is in a good agreement with the findings reported by Costantino and Bréon (2013).

Consistent with previous findings, we found that the CDR initially decreases with increasing AOD, followed by an increase after AOD reaches a value of 0.35. This result is consistent with Twomey's hypothesis that increasing aerosol abundance leads to more numerous but smaller cloud droplets at given constant cloud water content. The positive relation between CDR and AOD may be caused by microphysical processes, which is coupled with intense vapour competition and evaporation of smaller droplets as a result of a high abundance of aerosol particles. Also, the analysis of the variation of CF with increasing AOD showed that CF varies with AOD in a way similar to that of CDR. This finding differs from those by Koren et al. (2008) and Small et al. (2011) who observed that an increase in the cloud cover with an increasing AOD, followed by a decrease with higher AOD. COT was found to decrease with an increasing AOD. We argue that the radiative effect and retrieval artefact due to absorbing aerosol might be important factors in determining this relationship. This effect can result in increased cloud evaporation and reduced cloud cover. Meanwhile, CTP tends to increase as aerosol abundance increases, indicating that the aerosol is prone to expand the horizontal extension. In other words, we found that for well-mixed clouds over the YRD, the CDR becomes smaller with the increase of AOD in moderately polluted conditions which in principles in line with the Twomey effect, yet, the cloud fraction indicates a weak decrease which could be attributed only to the weak influence of evaporation caused by absorption of aerosols.

On the other hand, in polluted and heavily polluted conditions, a reduced cloud coverage can result in more solar radiation reaching the surface, causing surface heating and thus raises the surface temperature, which then destabilizes the atmosphere. The resulting advection transports water vapour from the surface to higher levels in the atmosphere, therefore producing more cloud. Meanwhile, CDR becomes larger as a result of the stronger water vapour competition in polluted and heavily polluted conditions. The COT decreases with the increasing values of AOD throughout the AOD range due to the radiative effect and possible retrieval artefacts. The behaviour of CTP is consistent with that of

COT, with the cloud getting thinner but with larger cover, so that CTP becomes larger with an increasing AOD.

Furthermore, joint correlative analysis of different aerosol and cloud properties revealed that smoke aerosols have a stronger impact on aerosol-cloud interaction due to their stronger absorption of solar radiation compared with polluted dust. Therefore, we can conclude that absorbing aerosols play an important role in the aerosol cloud interaction.

Constrained by relative humidity and boundary thermodynamic and dynamic conditions, the variation of cloud properties in response to aerosol abundance was analysed. In general, a high relative humidity can promote the formation of larger cloud droplets and expand cloud formation, irrespective of the vertical or horizontal level. With regard to LTS, stable atmospheric conditions can enhance the cloud cover horizontally. However, unstable atmospheric conditions can be helpful for the formation of thicker and higher clouds. Dynamically, an upward motion of air parcels can also facilitate the formation of thicker and higher clouds. Besides the meteorological controls mentioned above, other factors may be important in generating relations between aerosol and cloud properties, such as temperature advection. These results suggest that effects of ambient meteorological environments need to be considered when exploring the aerosol indirect effect. In summary, this study will greatly help us to understand the mechanisms of aerosol-cloud interaction and ultimately of aerosol indirect effects over the YRD.

**Acknowledgements**

[revised manuscript text omitted]